# Vertical profiles of black carbon measured by a micro-aethalometer in summer in the North China Plain

L. Ran[1], Z. Z. Deng[1], X. B. Xu[2], P. Yan[3], W. L. Lin[3], Y. Wang[2], P. Tian[4], P. C. Wang[1], W. L. Pan[1], D. R. Lu[1]

[1]Key Laboratory of Middle Atmosphere and Global Environment Observation, Institute of Atmospheric Physics, Chinese Academy of Sciences, Beijing, 100029, China
[2]Key Laboratory for Atmospheric Chemistry, Institute of Atmospheric Composition, Chinese Academy of Meteorological Sciences, Beijing, 100081, China
[3]Meteorological Observation Center, China Meteorological Administration, Beijing, 100081, China
[4]Beijing Weather Modification Office, Beijing, 100089, China

*Correspondence to*: L. Ran (shirleyrl@mail.iap.ac.cn) and Z. Z. Deng (dengzz@mail.iap.ac.cn)

**Abstract.** Black carbon (BC) is a dominant absorber in visible spectrum and a potent factor in climatic effects. Vertical profiles of BC were measured using a micro-aethalometer attached to a tethered balloon during the Vertical Observations of trace Gases and Aerosols (VOGA) field campaign, in summer 2014 at a semirural site in the North China Plain (NCP). The diurnal cycle of BC vertical distributions following the evolution of the mixing layer was investigated for the first time in the NCP region. Statistical parameters including identified mixing height ($H_m$) and average BC mass concentrations within the mixing layer ($C_m$) and in free troposphere ($C_f$) were obtained for a selected dataset of 67 vertical profiles. $H_m$ was usually lower than 0.2 km in the early morning and rapidly rose thereafter due to strengthened turbulence. The maximum height of the ML was reached in late afternoon. The top of a full developed ML exceeded 1 km on sunny days in summer, while stayed much lower on cloudy days. The sunset triggered the collapse of the ML and a stable nocturnal boundary layer (NBL) gradually formed. Accordingly, the highest level $C_m$ was found in the early morning and the lowest in the afternoon. In the daytime, BC almost uniformly distributed within the ML and significantly decreased above the ML. During the field campaign, $C_m$ averaged about 5.16±2.49 µg m$^{-3}$, with a range of 1.12 to 14.49 µg m$^{-3}$, comparable with observational results in many polluted urban areas such as Milan in Italy and Shanghai in China. As evening approached, BC gradually built up near the surface and exponentially declined with height. In contrast to the large variability found both in $H_m$ and $C_m$, $C_f$ stayed relatively unaffected through the day. $C_f$ was less than 10% of the ground level under clean conditions, while amounted to half of the ground level in some polluted cases. In-situ measurements of BC vertical profiles would hopefully have an important implication for accurately estimating direct radiative forcing by BC and improving the retrieval of aerosol optical properties by remote sensing in this region.

# 1 Introduction

Black carbon (BC), produced from incomplete combustion processes, is a strongly absorbing constituent of atmospheric aerosols (Moosmüller et al., 2009; Bond et al., 2013). As a major absorber in the visible spectrum, BC heats the atmosphere and largely counterbalances cooling effects of scattering aerosols on climate (Jacobson, 2001; Ramanathan et al., 2005; Stier et al., 2007). Another reason for BC to be publicly concerned is that inhaled BC poses a huge threat to human health (Janssen et al., 2012; Nichols, 2013).

Despite the significance of evaluating radiative forcing by BC, large uncertainties arise from limitations of current knowledge on emissions, distributions and physical properties of BC (Andreae, 2001; Streets, et al., 2001; Bond et al., 2006; IPCC, 2013). One critical aspect pertinent to climate response of BC is a high sensitivity of BC radiative impact to its vertical distributions (Zarzycki and Bond, 2010; Ban-Weiss et al., 2012; Samset et al., 2013). The importance of BC vertical distributions to the evolution of planetary boundary layer (PBL) and cloud properties has also been demonstrated by previous studies (Yu et al., 2002; Ramanathan and Carmichael, 2008; Ferrero et al., 2014). Nevertheless, vertical profiles of BC or aerosol absorption have only been scarcely measured in a few field campaigns (Safai et al., 2012 and references therein; Ryerson et al., 2013). Available information on BC vertical distributions are particularly limited in China (Zhang et al., 2012; Li et al., 2015; Zhao et al., 2015), issuing a challenge to reliably estimate regional climatic effects of BC under severe air pollution due to rapid economic growth and urbanization in this region (Menon et al., 2002; Liao and Shang, 2015 and references therein).

Platforms normally utilized to perform BC profiling are tethered balloons, aircrafts and unmanned aerial vehicles. BC vertical profiles obtained from in-situ measurements using tethered balloons are highly vertically resolved, revealing details within about 1 km above the ground, especially in the thin surface layer that is vital for human beings and where various sources are located (Ferrero et al., 2011a, Babu et al., 2011a; Li et al., 2015). Comparatively, aircrafts (Tripathi et al., 2005, 2007; Metcalf et al., 2012; Zhao et al., 2015) and unmanned aerial vehicles (Corrigan et al., 2008; Höpner et al., 2016) are more expensive, although they have advantages of reaching higher altitudes and for aircrafts more onboard instruments in the size and weight unable to be carried by tethered balloons and unmanned aerial vehicles. Fast flight speeds of these two platforms also compromise their spatial resolutions. In addition, high altitude balloon was employed to measure BC vertical profiles at high altitudes in free troposphere, e. g., two BC polluted layers found at about 4.5 km and 8 km altitudes were reported by Babu et al. (2011b). Indirect methods such as recently proposed Lidar remote sensing might be able to conduct continuous measurements (Miffre et al., 2015), they are however less accurate than in-situ measurements.

To advance understanding in impacts of atmospheric components including trace gases and aerosols on atmospheric environment and climate, an intensive field campaign, Vertical Observations of trace Gases and Aerosols (VOGA), was carried out in summer 2014 at a semirural site in the North China Plain (NCP), one of the most overcrowded and polluted regions in the world (Shao et al., 2006; Xu et al., 2011; Ma et al., 2011; Chen et al., 2012). A tethered balloon system equipped with instruments was employed for high vertical resolution measurements within 1 km above the ground. In this

study, we present results from in-situ measurements of BC vertical profiles using a lightweight (about 280 g) and small-sized (117 mm×66 mm×38 mm) micro-aethalometer (microAeth® Model AE-51, Magee Scientific, USA). The diurnal cycle of BC vertical distributions was explored for the first time in the NCP region.

## 2 Experiment

### 2.1 The site

The VOGA 2014 field campaign was carried out in the period from June 21 to July 14 at a semirural site Raoyang (38°14'N, 115°44'E, 20 m a.s.l.), about 50 km north of the city of Hengshui in the central NCP (Fig. 1). The county of Raoyang, in which the site is located, is less industrialized and relies mainly on agriculture, but surrounded by a cluster of industrial and populated cities within a distance of 100 km, and respectively about 190 km and 160 km southwest of the twin megacities of Beijing and Tianjin.

The spatial distribution of average aerosol optical depth (AOD) at 550 nm acquired from the level 2 version of the Moderate Resolution Imaging Spectroradiometer (MODIS) data (Levy and Hsu, 2015) is also displayed in Fig. 1. The feature of severe regional aerosol pollution with clearly defined pollution centers could be recognized from the AOD distributions in different months of the year 2014. The level of AOD in Raoyang could well represent regional aerosol conditions in NCP in July 2014, the month when most of tethered balloon flights took place (Fig. 1c). The AOD distributions could also cast some light on the seasonal variation of ambient aerosols in the area where launches of the tethered balloon were carried out, since measurements of absorbing aerosols are unavailable to obtain the temporal variation in that area. However, it should be noted that the seasonal cycle of ambient aerosols depends on both the aerosol loading and the relative humidity. As a consequence, it is complicated to draw a definite conclusion only from the AOD dataset about to what extent reported BC concentrations in this study represent the spatiotemporal variability in the area. A further examination on BC emissions (0.25°×0.25°) from four sectors (industry, power, residential activity, and transportation) in 2012 in the NCP region was performed on basis of emission data generated from the multi-resolution emission inventory for China (MEIC, http://www.meicmodel.org) model (Li et al., 2015), which is developed by a technology-based approach (Lei et al., 2011). It could also tell from the emission inventory of BC, that aerosol conditions at the site of Raoyang well represents the regional situation, with influences from several nearby emission centers (Fig. S1).

Moreover, it was found that another semirural site Gucheng, about 90 km north of Raoyang, shared a similarity with Raoyang in the AOD level as well as BC emissions. Seasonal and diurnal variations of surface BC mass concentrations in Gucheng were analyzed on basis of about six-year measurements (from 2006 January to 2012 July with the data completeness of 77.5%) using an aethalometer (Model AE-31, Magee Scientific, USA) with a temporal resolution of 5 min (Zhang et al., 2015). BC mass concentrations averaged about 9.6±8.4 µg m$^{-3}$ during 2006 and 2012 in Gucheng, with a lower level in summer and spring. The diurnal cycle of BC mass concentrations indicated higher values at night and a low valley in

the afternoon. Thus, it might be expected that BC mass concentrations measured in summer at Raoyang were probably lower than that in other seasons.

## 2.2 Instruments

A micro-aethalometer to measure aerosol absorption at 880 nm and a radiosonde to measure meteorological parameters (pressure, temperature, relative humidity, wind speed and direction) were attached to a helium-filled tethered balloon, with a volume of 30 m$^3$ and a payload weight of 10 kg. The fish-shaped balloon was well balanced during launches, thus the disturbance imposed on instruments by ascents and descents of the balloon could be largely minimized. Launches were scheduled every 3 hours between 06:00 and 21:00 (Local Time, LT) from the ground to 1 km height. However, feasible launches actually depended upon wind speed and the precipitation, as well as air traffic control. In total, 48 successful launches were conducted on 15 days, of which 40 reached above 0.5 km. Through the entire field campaign, 89 vertical profiles of BC were reliably obtained. Lack of data for several ascending or descending processes was primarily caused by discarding invalid data under wind gusts, which led to the violent swing of the tethered balloon and poor data quality. The balloon ascended at a rate of 1 m s$^{-1}$ and descended at a rate of 0.5 m s$^{-1}$ under the control of an electric winch. With a sampling interval of 1 s for all instruments, the observational vertical resolution was ~1 m. For analysis in this study, 20-m averaged data were used.

AE-51 operated with a similar principle of the aethalometer as described in Hansen et al. (1984). The intensity of transmitted light through a 3-mm diameter sample spot ($I$) and an aerosol-free reference spot ($I_0$) on a T60 Teflon-coated borosilicate glass fiber filter were simultaneously detected to obtain attenuation coefficients at the wavelength of 880 nm ($\sigma_{AE-51,880nm}$) following:

$$\sigma_{AE-51,880nm} = \frac{A \times \Delta ATN}{100 \times F \times \Delta t}, \tag{1}$$

where $A$ is the area of the aerosol collecting spot (0.071 cm$^2$), $F$ is the flow rate (150 ml min$^{-1}$), ATN denotes light attenuation and calculated from $100 \times \ln(I_0/I)$, $\Delta ATN$ is the change of ATN during the sampling interval $\Delta t$ (1 s). A homemade silica gel dryer was placed in front of the inlet to efficiently dry ambient aerosols. A test on the dryer was performed before the campaign to ensure the duration and efficiency of its usage.

## 2.3 Data processing

### 2.3.1 The smoothing algorithm

At a high temporal resolution of 1 s, steady increase of ATN with sampling time was not as usually found due to instrumental noises. Acquired data was consisted of many large values with positive or negative signs. A post-processing method, Optimized Noise-reduction Averaging (ONA) algorithm, has been developed by Hagler et al. (2011), where adaptive time-averaging of the BC data was conducted with the time window of averaging optimally chosen by $\Delta ATN$. In

this study, data dispersion due to high temporal resolution was treated by a new smoothing algorithm, Fluctuation Minimizing Smoothing (FMS). Similar to the ONA method, the FMS approach is also principally based upon the physical behavior of measured ATN. Despite that BC values determined from fluctuated ATN might drastically vary, large positive/negative pairs of BC values would always be found and counterbalance each other within a few seconds. Therefore, the FMS approach was devised to find pairs of BC values that differ largely with each other within a few seconds and make a compromise. The procedure was performed as following:

(1) For each data point $x_i$ of a continuously measured time series with $N$ records, calculate the difference between $x_i$ and all other data points within $n$ seconds to generate a matrix of $D_{i,j}=|x_i-x_{i+j}|$, where $i=1,2,…,N-1,N$ and $j=1,2,…,n-1,n$;

(2) Find the largest value of the matrix $D_{k,l}$, which is the difference between point $x_k$ and $x_{k+l}$;

(3) Replace $x_k$ and $x_{k+l}$ with $(x_k+x_{k+l})/2$ and set $D_{k,l}$ to a minus number;

(4) Replace any $D_{i,j}$ that has been calculated using $x_k$ or $x_{k+l}$, except for $D_{i,j}$ with a minus number. This step is optional, since it largely raises the computational cost;

(5) Repeat step (2)-(4) until no positive numbers in the matrix;

(6) Repeat step (1)-(5) for $m$ times to obtain acceptable smoothed data.

The smoothing window $n$ and the smoothing count $m$ were empirically chosen during the calculation. It should be kept in mind that using improper large $n$ or $m$ might wipe off some natural variations, although it will always give a smoother result. $n$ should be set to no more than 5, given that data fluctuation is normally already compensated within 5 s. With $n$ to be 5 and $m$ to be 1, the average of the target point was mostly contributed from neighboring data points within about 11 seconds, according to a weight of 80%. This consequently led to a vertical resolution of about 22 m for the ascent and 11 m for the descent after smoothing. Similarly, the vertical resolution was about 50~60 m for the ascent and 25~30 m for the descent when $m$ was set to be 5. Different choices of $m$ gave a similar pattern of vertical profiles, but with some differences in details. In this study, $m$ was set to be 5 to achieve a better smoothing for further calculations, though this caused a loss of the vertical resolution more than twice as large as that when just smoothing once. A comparison was made between unsmoothed data, smoothed data using the FMS approach in this study and the ONA method in Hagler et al. (2011), as well as 20-m averaged data using those two algorithms. It was found that both algorithms could properly deal with data fluctuation caused by instrumental noises without introducing artificial features (e.g., Fig. 3a-3c). However, the FMS method seemed to be more capable of reliably removing outliers in some cases (e.g., Fig. 3d-3f). The comparison indicated that the FMS procedure could effectively reduce data fluctuation while still preserve reasonable variability of the profile.

### 2.3.2 The correction method

Measured $\sigma_{AE-51,880nm}$ suffered from systematic biases introduced by the filter-based technique. In order to determine BC absorption coefficients ($\sigma_{BC}$) from $\sigma_{AE-51,880nm}$, corrections were required to tackle with three types of artifacts. The shadowing effect, an artifact that results in gradual artificial reduction in $\sigma_{AE-51,880nm}$ due to the saturation of the filter with increasing aerosol loading, leads to an underestimation of $\sigma_{BC}$ and a discontinuity after changing to a new sample spot

(Weingartner et al., 2003). Various methods have been developed to address the shadowing effect (Weingartner et al., 2003; Virkkula et al., 2007; Ran et al., 2016). However, this artifact could be neglected in this study, since no ATN exceeded 20 with a new filter for each launch. The other two artifacts cause an overestimation of $\sigma_{BC}$ by enhancing light attenuation, either due to aerosol scattering or the multiple scattering of filter fibers (Weingartner et al., 2003; Arnott et al., 2005; Schmid et al., 2006; Collaud Coen et al., 2010). A correction factor ($C$) was needed to correct these two artifacts.

The $C$ factor was derived from a surface comparative test for about 1 week in Beijing among AE-51, a 7-wavelength aethalometer (Model AE-31, Magee Scientific, USA) and a multi-angle absorption photometer (MAAP, Model 5012, Thermo, USA). Continuous operation of AE-51 was carried out in order to obtain a proper size of the dataset in limited time period, despite that AE-51 is in principle not designed to be operated around the clock. A daily check of the flow rate was performed using a Gilibrator-2 Diagnostic Kit (Sensidyne, USA) to ensure the stability of the flow and thus data quality.

AE-31 suffered instrumental artifacts in the same way as AE-51. Details of the correction scheme developed for tackling with AE-31 artifacts were described in Ran et al. (2016). Briefly, the correction scheme combined the modified Virkkula method (Virkkula et al., 2007) to treat the shadowing effect and the Schmid method (Schmid et al., 2006) to treat filter multiple scattering and aerosol scattering effects. The modified Virkkula method assumed a linear relationship of BC mass concentrations and time across the filter change, particularly, a quadratic relationship for special cases where ambient BC experienced a peak-shaped variation, instead of constant BC mass concentrations as in Virkkula et al. (2007). The wavelength-dependent correction factor ($C_\lambda$) could be obtained following procedures in Schmid et al. (2006). The temporal resolution of AE-31 during the comparative test was 2 min.

MAAP continuously measured aerosol absorption coefficients at the actual wavelength of 637 nm ($\sigma_{MAAP,637nm}$) instead of the nominal wavelength of 670 nm (Müller et al., 2011), with a temporal resolution of 1 min. Multi-angle detections of the transmittance and reflectance, as well as a two-stream approximation in the radiative transfer scheme adopted by MAAP have significantly improved measurements of ambient aerosol absorption coefficients (Petzold and Schönlinner, 2004; Petzold et al., 2005). Hyvärinen et al. (2013) reported an artifact of underestimated BC mass concentrations after a spot change and attributed it to yet unconfirmed causes as erroneous dark counts in the transmitted light photodetector and an instrument internal averaging procedure of the photodetector raw signals. It was stated that this artifact could be observed with a BC mass accumulation rate, as the product of BC mass concentrations and the flow rate of MAAP, larger than 0.04 $\mu g \, min^{-1}$, which corresponds to 3 $\mu g \, m^{-3}$ at the flow rate of 1 $m^3 \, h^{-1}$. However, no apparent underestimation of BC mass concentrations was found in this study, even for cases where BC mass concentrations exceeded 8 ug $m^{-3}$ (Fig. S2). Consequently, measured $\sigma_{MAAP,637nm}$ without any corrections were used for subsequent calculations and taken as real values.

Three steps were taken to obtain the $C$ factor. Firstly, aerosol absorption Angström exponent ($\alpha$) over the spectrum span of 660 and 880 nm was derived from absorption coefficients $\sigma_{AE-31,660nm}$ and $\sigma_{AE-31,880nm}$, which were corrected from attenuation coefficients at 660 and 880 nm measured by AE-31. Hence, $\alpha$ was calculated from:

$$\alpha = \frac{\ln(\sigma_{AE-31,660nm}) - \ln(\sigma_{AE-31,880nm})}{\ln(880) - \ln(660)}. \tag{2}$$

Secondly, $\alpha$ for the spectrum of 660 and 880 nm was used to represent $\alpha$ over the span of 637 and 880 nm. Therefore, $\sigma_{MAAP,880nm}$ was quantified from measured $\sigma_{MAAP,637nm}$ following the spectral dependence of aerosol absorption coefficients in the form of $\lambda^{-\alpha}$:

$$\sigma_{MAAP,880nm} = \sigma_{MAAP,637nm} \times (\frac{880}{637})^{-\alpha} ,$$ (3)

Finally, reduced major axis regression of attenuation coefficients $\sigma_{AE-51,880nm}$ (ATN<10) measured by AE-51 and absorption coefficients $\sigma_{MAAP,880nm}$ calculated from MAAP and AE-31 yielded the $C$ factor of 2.98±0.05 with 95% confidence (Fig. 2). It was noted that the $C$ factor for AE-51 was reported as 2.05±0.03 with 95% confidence in Ferrero et al. (2011a). Possible explanations on such a difference in the $C$ factor might be found in aerosol chemical compositions in the NCP region and the Po Valley basin. Besides, the $C$ factor in Ferrero et al. (2011a) was obtained from Mie calculations, and thus was subject to uncertainties resulting from assumptions such as BC size distributions, BC mixing state and particle morphology. In addition, the choice of the correction scheme for AE-31 measurements in this study might introduce uncertainties to $\alpha$ and thereby $\sigma_{MAAP,880nm}$. Using a constant $C$ factor for AE-31 as also often used in some studies (e.g., Weingartner et al., 2003; Sandradewi et al., 2008) instead of the wavelength-dependent $C_\lambda$ results in an underestimation of $\alpha$ over the 660-880 nm spectrum by about 19.5%. This consequently leads to an overestimation of $\sigma_{MAAP,880nm}$ and the $C$ factor for AE-51 by about 9.6% and 8.4%, respectively. By dividing $C$ into $\sigma_{AE-51,880nm}$, absorption coefficients $\sigma_{BC}$ could be estimated. As for the mass concentration of BC ($m_{BC}$), it could be directly converted from $\sigma_{AE-51,880nm}$ using the documented 'specific attenuation' of 12.5 $m^2 g^{-1}$, considering that BC is the major contributor to aerosol absorption at 880 nm (Ramachandran and Rajesh, 2007).

### 2.3.3 The calculation of meteorological parameters

Instead of measured temperature ($T$) and relative humidity (RH), two conservative quantities, potential temperature ($\theta$) and specific humidity ($q$) were used to provide information on the evolution of PBL. The calculation of $\theta$ (K) followed:

$$\theta = T \times \left( p_0 / p \right)^{R_d / C_{pd}} ,$$ (4)

where $p$ and $p_0$ are respectively measured pressure and the standard pressure at the sea level (hPa), $T$ is measured temperature (K), $R_d$ is the individual gas constant for dry air (287 J kg$^{-1}$ K$^{-1}$), $C_{pd}$ is the specific heat at constant pressure for dry air at the standard temperature (1005 J kg$^{-1}$ K$^{-1}$). Calculated $\theta$ was then expressed in degree Celsius ($^o$C) in order to be straightforward. Specific humidity ($q$, g kg$^{-1}$) was calculated from:

$$q = \varepsilon \times \frac{e}{p - (1-\varepsilon) \times e} \times 1000 ,$$ (5)

where $\varepsilon$ is the ratio of the molecular weight of water vapour to that of dry air (0.622), $e$ is the vapour pressure of water (hPa), which is the product of measured RH (%) and calculated water saturation pressure ($e_s$) at each measured temperature $T$.

## 3 Results and Discussion

### 3.1 Vertical distributions of BC and meteorological parameters

A set of 67 vertical profiles of BC and meteorological parameters were selected. The mixing layer (ML) could be clearly discerned for profiles measured in the daytime. Exceptions were those measured around noon, when the top of the ML, often higher than 1 km in summer, was beyond reach of the tethered balloon. In addition, several launches took place in the evening to observe BC vertical distributions shaped by the nocturnal boundary layer (NBL). A statistical summary of the field campaign and the meteorology is given in Table 1. Most profiles were sampled under mild winds in the morning and evening, when relatively stable conditions suitable for launches of the tethered balloon were more easily encountered. Air masses were mainly carried, either by southeasterly or southwesterly winds, from areas densely populated and heavily industrialized, also ramified with railways and highroads.

Figure 4 illustrates an example of vertical distributions of BC, $\theta$, $q$, and winds during one launch on July 8, 2014 (10:41-11:27 LT). 20-m averaged data are denoted by dots in dark color for the ascent and light color for the descent. Arrows depict vector mean horizontal winds, with the length to represent wind speed and the arrowhead to give wind direction following meteorological definitions. The height of a clearly defined ML was marked by a solid line for each BC vertical profile. A nearly uniform distribution of BC, as well as $\theta$ and $q$, was observed within the strongly convective ML. Large vertical gradient of $m_{BC}$ across entrainment layer (EL) led to approximately 50% reduction from the level in the ML to that in free troposphere (FT). Concomitantly, a substantial reduction in $q$, whereas an increase in $\theta$, was also immediately found above the ML. The difference in profiles between the ascent and the descent evidently indicated a rapid evolution of the ML near noon and its impact on vertical distributions of BC and meteorology. More discussions regarding diurnal variations of BC vertical profiles will be found in next section.

The mixing height could be determined by applying the gradient method to the entire dataset (Seibert et al., 2000; Kim, et al., 2007). Generally, the mixing height determined from profiles of $m_{BC}$ ($H_{m,BC,gradient}$) agreed well with that from profiles of $\theta$ ($H_{m,\theta}$) and $q$ ($H_{m,q}$) as shown in Fig. S3. The reliability of estimating the mixing height from vertical measurements of BC had also been demonstrated in previous studies (Ferrero et al., 2010; Ferrero et al., 2011b). On the other hand, typical daytime profiles of $m_{BC}$ could be well characterized by the sigmoid function:

$$m_{BC} = C_{ms} - \frac{C_{ms} - C_{fs}}{e^{-(h-H_{m,BC,sigmoid})/H_e} + 1},\tag{6}$$

where $C_{ms}$ and $C_{fs}$ are respectively characteristic $m_{BC}$ within the ML and in FT, $H_{m,BC,sigmoid}$ is the mixing height derived from BC vertical profiles using the sigmoid function, $H_e$ represents the thickness of the EL, $h$ is the height at which each 20-m averaged $m_{BC}$ is obtained. The parameters $C_{ms}$, $C_{fs}$, $H_{m,BC,sigmoid}$ and $H_e$ could be directly determined by fitting measured $m_{BC}$ at each height $h$ using Eq. (6) as shown by the example (Fig. S4). A comparison was made between $H_{m,BC,gradient}$ and $H_{m,BC,sigmoid}$ for typical daytime BC vertical profiles. Results from the two methods agreed quite well with each other, with a difference of less than 2 % (Fig. S5). In addition to reliably estimating the mixing height as the gradient method, the sigmoid

function could also directly determine parameters including $C_{ms}$, $C_{fs}$, and $H_e$. Therefore, the sigmoid function was chosen to obtain all parameters for typical daytime BC profiles. As for non-typical daytime BC profiles, either as a result of a polluted layer in FT or barely sufficient height into FT reached by the tethered balloon, a sigmoid function was not applicable and the gradient method was used to estimate the mixing height. Near sunset, the NBL forms along with the collapse of the ML

(Kaimal and Finnigan, 1994; Lazaridis, 2011). The structure of the NBL is more complicated and the determination of the NBL height bears more uncertainty (Yu, 1978; Poulos, et al., 2002). A subjective procedure based on the gradient method was utilized to estimate the NBL height. In summary, the mixing height determined from vertical profiles of BC, either by the gradient method or the sigmoid function, as well as the estimated NBL height, were denoted as $H_m$ in this study for convenience.

Table 2 summaries calculated parameters for BC vertical profiles during different periods. Average $m_{BC}$ within the ML and in FT were denoted as $C_m$ and $C_f$, which were $C_{ms}$ and $C_{fs}$ for typical daytime BC profiles. As for non-typical daytime BC profiles, $C_m$ was the average of $m_{BC}$ below $H_m$. When $H_m$ exceeded 1 km, $m_{BC}$ measured along the ascending/descending path from the ground to the maximum height reached by the tethered balloon were averaged to obtain $C_m$. For NBL cases, only $C_f$ was obtained, by averaging $m_{BC}$ above $H_m$. A large variability was found in $C_m$, with a range of 1.12 to 14.49 µg m$^{-3}$.

On average, $C_f$ stayed relatively steadily, though it ranged from 0.36 µg m$^{-3}$ in clean cases to 3.14 µg m$^{-3}$ under polluted conditions.

Statistically, vertical profiles of BC were categorized into two types, according to their shapes along the normalized height ($H_{Nor}$), which was calculated from $h/H_m-1$ (Ferrero et al., 2014). $H_{Nor}$ shows the surface with a value of -1 and the top of the ML with a value of 0. One type of BC vertical profiles, exclusively measured in the daytime, displayed a vertical distribution

typically shaped by a well-mixed mixing layer (Fig. 5a). BC almost uniformly distributed within the ML, in a similar way as the example (Fig. 4), with an average $m_{BC}$ level of about 5.16±2.49 µg m$^{-3}$, or $\sigma_{BC}$ of about 21.64±10.45 Mm$^{-1}$. This is comparable with what has been observed within the PBL in many polluted urban areas (Table 3). A couple of highly polluted cases, where $C_m$ exceeded 10 µg m$^{-3}$, were observed in the early morning. The formation of the convective ML just started at that time and vertical turbulence was yet not strong enough to dilute the high level of BC, which was primarily

emitted near the ground and accumulated within the stable NBL over night. Due to a sharp reduction in $m_{BC}$ from the ML to FT, average $C_f$ was about 1.61±0.94 µg m$^{-3}$. The second type of BC vertical profiles revealed the feature of trapped air pollutants in the shallow stable NBL. For each BC profile (grey lines in Fig. 5b), $m_{BC}$ nearly exponentially declined with $H_{Nor}$, as a result of weakened turbulence and vertical dispersion. Under polluted conditions, $C_f$ could reach as high as 2.83 µg m$^{-3}$, otherwise usually well below 1 µg m$^{-3}$.

**3.2 Diurnal variations of BC vertical profiles**

An investigation on diurnal variations of BC vertical profiles was able to be undertaken on July 1, 8, and 13, when measurements generally covered the period from early morning to afternoon or evening. On other observational days, the dataset was incomplete due to aborted launches under strong winds or precipitation. For clarity, only the vertical profile

during the descent of each launch is plotted in Fig. 6. Additionally, profiles not included into the selected dataset are also shown.

A distinct diurnal variation was found both in the shape of BC vertical profiles and the level of $m_{BC}$ along the profile. To a great extent, the diurnal cycle of BC vertical profiles followed the evolution of the ML. In the early morning, BC was largely restricted within a thin ML less than 0.2 km above the ground when vertical turbulence was still weak. $m_{BC}$ dropped dramatically from a high level within the ML to a significantly low level above the ML, even less than 10% as could be found on July 13. In polluted cases, however, $m_{BC}$ in FT could exceed 2 μg m$^{-3}$ and sometimes reach nearly 50% of that at the surface (e.g., profiles on July 1 and 8). This might imply the existence of a polluted residual layer above the stable surface layer formed after the sunset in previous evening, yet unable to be further discussed without continuous measurements from the day before. Also the characteristics of $m_{BC}$ in FT should be affected by the advection. After sunrise, vertical turbulence in the ML gradually strengthened as the earth was continuously heated by solar radiation. The convective ML developed quickly in the forenoon, with elevated $H_m$ and decreased $C_m$. Marked changes in BC vertical profiles could be observed even during one launch when the mixing height rose so rapidly, as shown in the example (Fig. 4). The maximum height of the ML commonly exceeded 1 km in the afternoon during the field campaign (e.g., profiles on July 8), except on cloudy days when the ML did not fully develop. As the evening approached, convection abruptly decayed and a stable NBL began to form. $m_{BC}$ in FT promptly declined to its typical level, often leaving no trail to indicate the daytime mixing with polluted air masses beneath. In contrast, air pollutants including BC quickly built up in the shallow NBL. Sometimes, a residual layer with a relatively high level of $m_{BC}$ (>2 μg m$^{-3}$) could be formed above the NBL where the remnant of the daytime mixing layer might be traced after its collapse (e.g., profiles on July 8). This would undoubtedly have an impact on measured $m_{BC}$ above the mixing layer on the next day, leading to a polluted background in FT (e.g., in the morning of July 1 and 8). The role of the residual layer in affecting the evolution of the PBL still stays controversial, though it has been consented that BC could heat the PBL and intensify atmospheric stability. Ding et al. (2016) demonstrated the importance of the "dome effect" of BC in the PBL especially the upper PBL, suppressing the PBL height and enhancing haze pollution within a lower PBL. Whereas in Zhang et al. (2012), a limited warming effect of BC in an elevated aerosol layer and limited induced increase in the strength of atmospheric inversion were indicated.

It was interestingly noticed that a polluted layer with a thickness of about 0.3 km, extending from the height of 0.2 km to 0.5 km, lay right above the ML in the early morning on July 1. The polluted layer could also be accordingly recognized from meteorological features of more moisture and wind shear across its boundaries. Wind direction was southeast in the ML and FT, while southwest in the polluted layer. Average $m_{BC}$ within the polluted layer was about 4 μg m$^{-3}$, half of that near the surface and double of that in FT. Actually, the vertical profile within the morning ML quite resembled characteristics of a nocturnal profile even though the sun had risen, suggesting that strong inversion of the NBL still overrode weak vertical turbulence at that time. As surface emissions from human activities kept increasing, $m_{BC}$ within the ML elevated. Further development of vertical convection broke the layered structure, leading to the merge of the polluted layer and the ML.

Slowly, the polluted layer vanished into the convective ML, though remained until 9:00 LT in the morning. An elevated aerosol layer (0.7~1 km), in which BC was well aged, has previously been observed in the afternoon over Beijing by Zhang et al. (2012), and was attributed to the residual nocturnal stable layer. Possible origin of the polluted layer present here and the status of BC in the layer were unable to be decided on basis of limited observations. Nonetheless, a higher $q$ in the polluted layer than at the surface and different wind direction might imply an elevated residual layer formed after sunset on the day before and transported from somewhere else. It was quite cloudy and sultry during the daytime, and a heavy rainfall occurred near the evening. Thus, the top of the ML was still below 0.5 km when the launch was carried out around noon. Despite that the tethered balloon failed to reach 1 km due to strong winds aloft, a part of a relatively polluted layer with apparent increase in moisture and potential temperature could be roughly distinguished. Wind direction in the polluted layer was southwest, whereas predominant wind direction beneath had changed to southeast. The polluted layer, extending from about 0.7 km to the top of the reachable height, disappeared during the last launch around 14:00 LT on that day and was probably a transported plume.

Figure 7 displays $H_m$ and $m_{BC}$ from the entire selected dataset on a diurnal scale. Observations on each day are shown in one color. Variations in $H_m$ through the field campaign generally resembled the diurnal cycle observed on an individual day (Fig. 7a). The ML, though still shallow, developed slowly after sunrise. In addition to a high level of BC trapped near the ground by the stable NBL, fresh emissions from daytime human activities further aggravated surface pollution. Observed $m_{BC}$ was usually quite high in the early morning, sometimes even experienced an increase with time, depending upon how fast the ML developed and emissions increased. The thickening of the mixing depth accelerated in late morning on sunny days, whereas impeded on cloudy days only to form a thin mixing layer. Accordingly, $C_m$ (solid dots in Fig. 7b) largely reduced from morning to noon in strongly convective ML on sunny days, however, stayed relatively stable through morning on cloudy days. Comparative cases could be found on July 1 and 14. Based on current knowledge of the boundary layer, the maximum height of the ML on sunny days would be reached in late afternoon, but could not be observed in this study since it was well above the flight limit of the tethered balloon. The collapse of the ML was not directly captured. However, intensive measurements favored by the meteorology on July 8 revealed that the convective ML quickly faded away on a time scale of 1 hour near sunset. Instead, a stable and shallow NBL confined nighttime emissions near the ground, giving rise to enhanced surface pollution (open dots in Fig. 7b).

## 4 Conclusions

In-situ measurements of black carbon (BC) vertical profiles were carried out at a semirural site during the VOGA 2014 summer field campaign, using a micro-aethalometer attached to a tethered balloon system. A set of 67 BC vertical profiles was reliably collected and diurnal variations of BC vertical profiles were examined.

Vertical distributions of BC and meteorological parameters, as well as the mixing height ($H_m$) identified from BC profiles, experienced a distinct diurnal cycle guided by the development of the mixing layer (ML). In the morning, weak turbulence

within the shallow ML (thinner than 0.2 km) favored the accumulation of enhanced surface emissions, sometimes leading to a severe pollution with BC mass concentrations ($m_{BC}$) exceeding 10 µg m$^{-3}$. The high level of BC within the ML was diluted with the fast development of a strongly convective ML. $H_m$ reached its maximum in late afternoon and exceeded 1 km on sunny days, whereas stayed much lower on cloudy days. Typically in the daytime, BC uniformly distributed within the ML and sharply decreased above the ML to a level even less than 10% of that near the surface. Average $m_{BC}$ within the ML ($C_m$) was about 5.16±2.49 µg m$^{-3}$ over the entire field campaign, with a range of 1.12 to 14.49 µg m$^{-3}$, comparable with what has been observed in many polluted urban areas. $C_f$, average $m_{BC}$ in free troposphere (FT), was below 1 µg m$^{-3}$ under clean conditions and amounted to half of the ground level under pollution conditions. In the evening, BC quickly built up near the surface and exponentially declined with height, following the collapse of the ML and the formation of a stable nocturnal boundary layer (NBL). Particularly, two polluted layers were observed at different time and heights on July 1. Origins of the polluted layers were not able to be evidently decided. However, the presence of BC polluted layers in FT, either as an elevated residual layer in the morning or a transported plume in the afternoon as suggested by the history of the layer and the meteorology, might considerably influence direct radiative forcing by BC and atmospheric stability.

## Acknowledgements

This research was funded by the National Natural Science Foundation of China (NSFC) under Grant No. 41305114, 41205098, 41330442 and 41127901. This work was also supported by China Special Fund for Meteorological Research in the Public Interest (No. GYHY201206015) and National Science and Technology Major Project (2016YFC0200403). We are grateful to Yong Wang, Chuncheng Ji, Qihua Du, Hanze Yu and Xinpan Li for their cooperation in launching the tethered balloon. We also thank Raoyang Meteorological Bureau for providing the location for our measurements. The Terra and Aqua/MODIS Aerosol Level 2 Products were acquired from the Level-1 & Atmosphere Archive and Distribution System (LAADS) Distributed Active Archive Center (DAAC), located in the Goddard Space Flight Center in Greenbelt, Maryland (https://ladsweb.nascom.nasa.gov/).

# Appendix A  Nomenclature Table

| Symbol | Definition |
| --- | --- |
| $\alpha$ | Aerosol absorption Ångström exponent, here calculated from $\sigma_{AE\text{-}31,660nm}$ and $\sigma_{AE\text{-}31,880nm}$ |
| $\sigma_{AE\text{-}31,660nm}$ | Aerosol absorption coefficient measured by AE-31 at the wavelength of 660 nm |
| $\sigma_{AE\text{-}31,880nm}$ | Aerosol absorption coefficient measured by AE-31 at the wavelength of 880 nm |
| $\sigma_{MAAP,670nm}$ | Aerosol absorption coefficient measured by MAAP at the wavelength of 670 nm |
| $\sigma_{MAAP,880nm}$ | Aerosol absorption coefficient at the wavelength of 880 nm, calculated from $\alpha$ and $\sigma_{MAAP,670nm}$ |
| $\sigma_{AE\text{-}51,880nm}$ | Aerosol absorption coefficient measured by AE-51 at the wavelength of 880 nm |
| $\sigma_{BC}$ | Absorption coefficient of black carbon at the wavelength of 880 nm |
| $\lambda$ | Wavelength |
| $\theta$ | Potential temperature |
| ATN | Light attenuation |
| $C$ | Correction factor for filter multiple scattering and aerosol scattering |
| $C_f$ | Average $m_{BC}$ in the free troposphere |
| $C_m$ | Average $m_{BC}$ within the mixing layer |
| $C_{fs}$ | Characteristic $m_{BC}$ in the free troposphere, derived from sigmoid fitting |
| $C_{ms}$ | Characteristic $m_{BC}$ within the mixing layer, derived from sigmoid fitting |
| $H_e$ | The thickness of the entrainment layer, derived from sigmoid fitting |
| $H_m$ | The height of the mixing layer, also used to denote the height of the NBL layer in this study |
| $H_{m,BC,gradient}$ | The height of the mixing layer estimated from the vertical profile of BC using the gradient method |
| $H_{m,BC,sigmoid}$ | The height of the mixing layer estimated from the vertical profile of BC using the sigmoid function |
| $H_{m,\theta}$ | The height of the mixing layer estimated from the vertical profile of $\theta$ using the gradient method |
| $H_{m,q}$ | The height of the mixing layer estimated from the vertical profile of $q$ using the gradient method |
| $H_{Nor}$ | The normalized height, defined as $h/H_m\text{-}1$ |
| $m_{BC}$ | Mass concentration of black carbon, converted from $\sigma_{AE\text{-}51,880nm}$ |
| $q$ | Specific humidity |
| $RH$ | Relative humidity |
| $T$ | Temperature |

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

Table 1 A statistical summary of the field campaign and the meteorology. $H_{max}$ denotes the maximum height reached by the tethered balloon. $T_{10m}$, $RH_{10m}$, and $WS_{10m}$ indicated temperature, relative humidity, and wind speed within 10 m near the surface, in the form of average value±standard deviation. $WD_{10m}$ represents frequently encountered wind direction. The last column gives the weather that mostly occurred during each period.

| Periods | Profiles | $H_{max}$ (km) | $T_{10m}$ (°C) | $RH_{10m}$ (%) | $WS_{10m}$ (m s$^{-1}$) | $WD_{10m}$ | Weather |
|---|---|---|---|---|---|---|---|
| 06:00-09:00 | 23 | 0.28~0.92 | 25.3±0.8 | 62.7±3.2 | 2.9±1.3 | Southwest | Cloudy |
| 09:00-12:00 | 20 | 0.32~1.06 | 26.0±1.5 | 61.0±3.4 | 2.6±1.1 | Southwest | Cloudy |
| 12:00-15:00 | 4 | 0.84~1.14 | 28.7±2.3 | 55.9±5.0 | 2.2±1.4 | Southeast | Fine |
| 15:00-18:00 | 6 | 1.04~1.08 | 28.5±2.4 | 58.7±4.6 | 2.1±1.1 | Southeast | Fine |
| 18:00-21:00 | 14 | 0.26~1.16 | 29.1±2.2 | 45.0±3.3 | 1.8±0.8 | Southwest | Fine |

Table 2 Statistical parameters for BC vertical profiles measured during different periods. Notations for the parameters have been given in the text.

| Parameters | 06:00-09:00 23 Profiles | 09:00-12:00 20 Profiles | 12:00-15:00 4 Profiles | 15:00-18:00 6 Profiles | 18:00-21:00 14 Profiles |
|---|---|---|---|---|---|
| $H_m$ (km) | 0.08~0.29 | 0.18~0.87 | 0.43~1.14 | 0.69~1.08 | 0.19~0.25 |
| $H_e$ (km) | 0.05±0.04 | 0.05±0.03 | | | |
| $C_m$ (µg m$^{-3}$) | 3.1~14.49 (6.61±2.93) | 2.56~5.80 (4.42±1.11) | 1.12~6.15 (3.77±2.06) | 2.04~4.11 (3.05±0.86) | |
| $C_f$ (µg m$^{-3}$) | 0.46~3.14 (1.67±0.95) | 0.41~2.51 (1.77±0.75) | | | 0.36~2.83 (1.36±1.12) |

Table 3 Measurements of BC vertical distributions in different regions. $m_{BC}$ is the range and/or average value±standard deviation of BC mass concentrations, depending upon what is available in the literature. Height specifies how $m_{BC}$ is obtained.

| Locations | Type | Period | Method | $m_{BC}$ (µg m$^{-3}$) | Height | References |
|---|---|---|---|---|---|---|
| Los Angeles Basin, USA | Mixture | May 2010 | Aircraft, SP2* | 0.01~0.53 | 0.3 km | Metcalf et al., 2012 |
|  |  |  |  | 0.03±0.05 | FT |  |
| Milan, Italy | Urban | Feb. 2010 | Tethered balloon, AE-51 | 7.57 ± 1.28 | ML | Ferrero et al., 2014 |
|  |  |  |  | 2.03 ± 0.34 | FT |  |
| Multi-cities, India | Urban | Jun. 2005 | Aircraft, AE-42, AE-21 | 1.50~7.50 | PBL | Safai et al., 2012 and references therein |
| Beijing, China | Urban | May-Jun. 2012 | Aircraft, SP2 | 0.24~4.02 | PBL | Zhao et al., 2015 |
| Shanghai, China | Urban | Dec. 2013 | Tethered balloon, AE-31 | 3.20~7.30 | 0-1 km | Li et al., 2015 |
| Central NCP, China | Semirural | Jun.-Jul. 2014 | Tethered balloon, AE-51 | 1.12~14.49 | ML | This study |
|  |  |  |  | (5.16±2.49) |  |  |
|  |  |  |  | 0.36~3.14 | FT |  |
|  |  |  |  | (1.61±0.94) |  |  |

\* SP2: Single Particle Soot Photometer

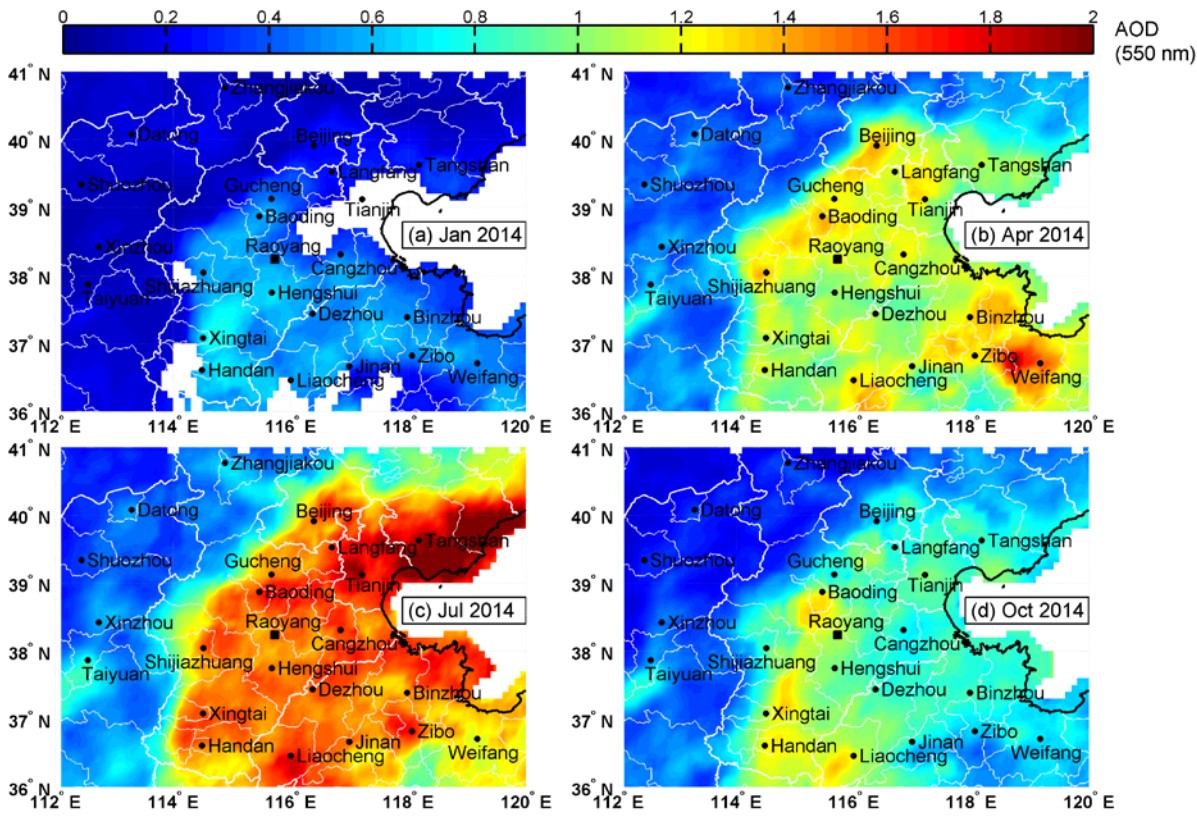

Figure 1. The spatial distribution of averaged MODIS aerosol optical depth (AOD) at 550 nm in (a) January; (b) April; (c) July; (d) October, 2014 in the NCP. The locations of the semirural site Raoyang and major cities are respectively marked by square and dots. Only grids with the fraction of valid data exceeding 30% in the month are shown.

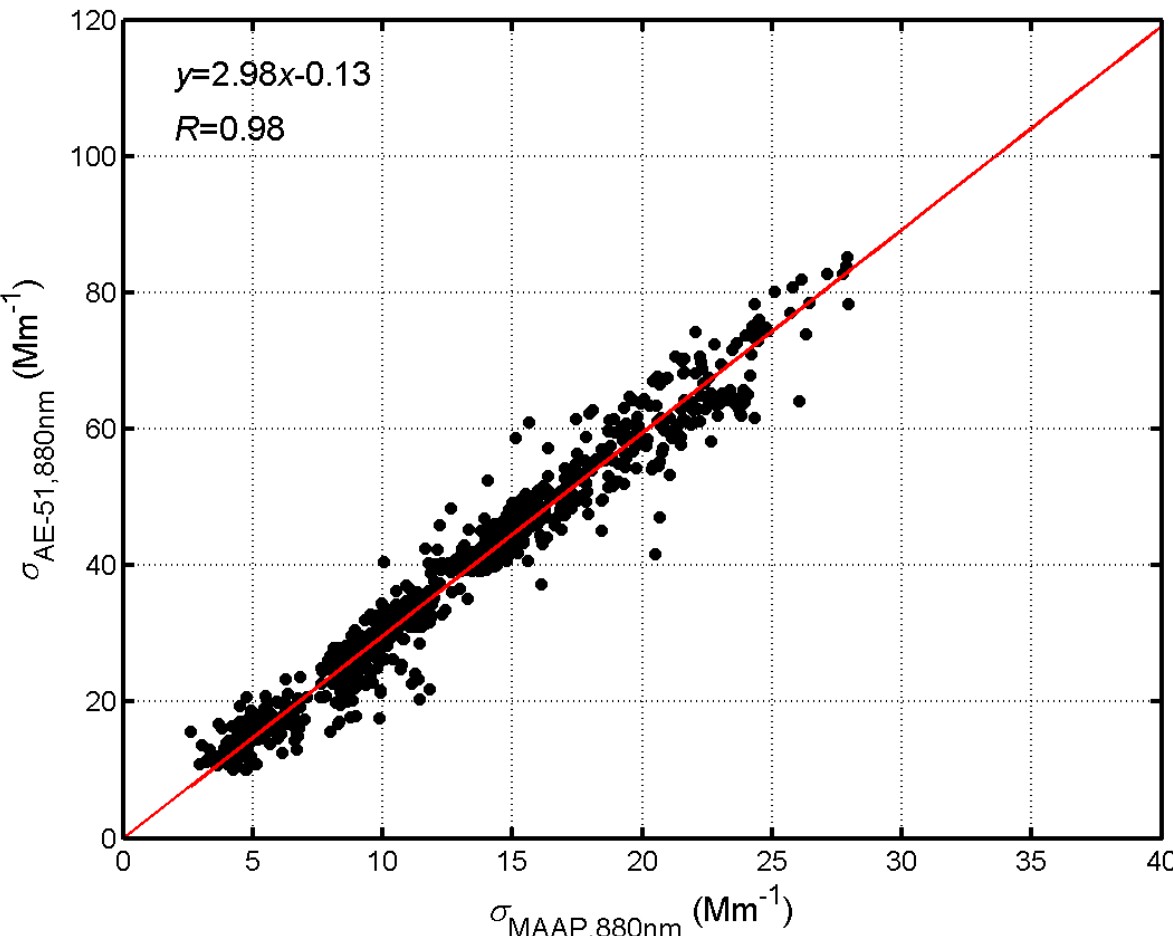

Figure 2. Reduced major axis regression of attenuation coefficients $\sigma_{\text{AE-51,880}}$ (ATN<10) measured by AE-51 and absorption coefficients $\sigma_{880nm}$ calculated from concomitant MAAP and AE-31 measurements in the comparative test.

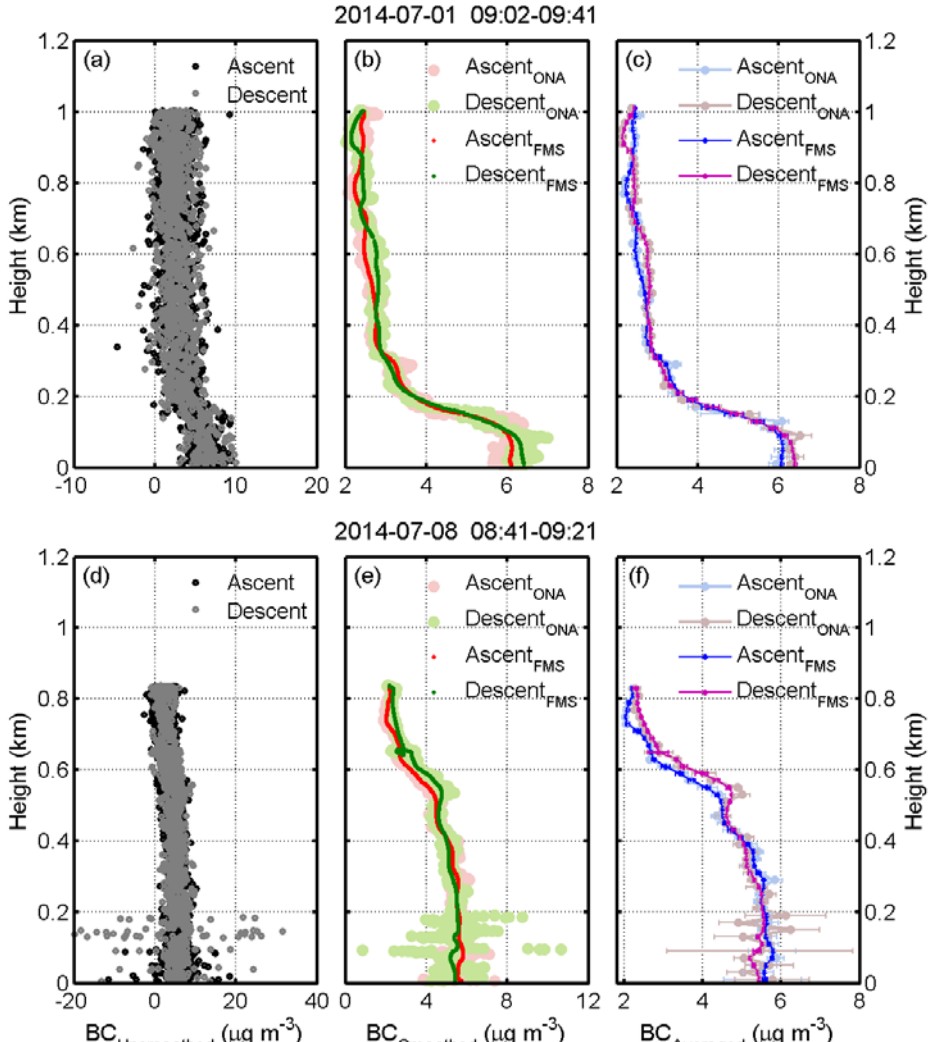

Figure 3. (a) Unsmoothed BC mass concentrations measured with a temporal resolution of 1 s on July 1, 2014 (09:02-09:41 LT). Data points collected from the ascending and descending process are respectively marked in black and grey dots. (b) Smoothed BC mass concentrations using two algorithms. Data points processed by the ONA method are displayed in large pink dots for the ascent and in light green color for the descent. Data points processed by the FMS method are denoted by small red dots for the ascent and green dots for the descent. (c) 20-m averaged profiles based upon smoothed data using two algorithms. Dots indicate 20-m averages, with standard deviations in error bars. Results from the ONA and FMS methods are respectively given in the color of light blue and blue for the ascent, while in the color of light purple and purple for the descent. (d)-(f) Measured and processed BC vertical profiles on July 8, 2014 (08:41-09:21 LT). The caption is the same as that in (a)-(c).

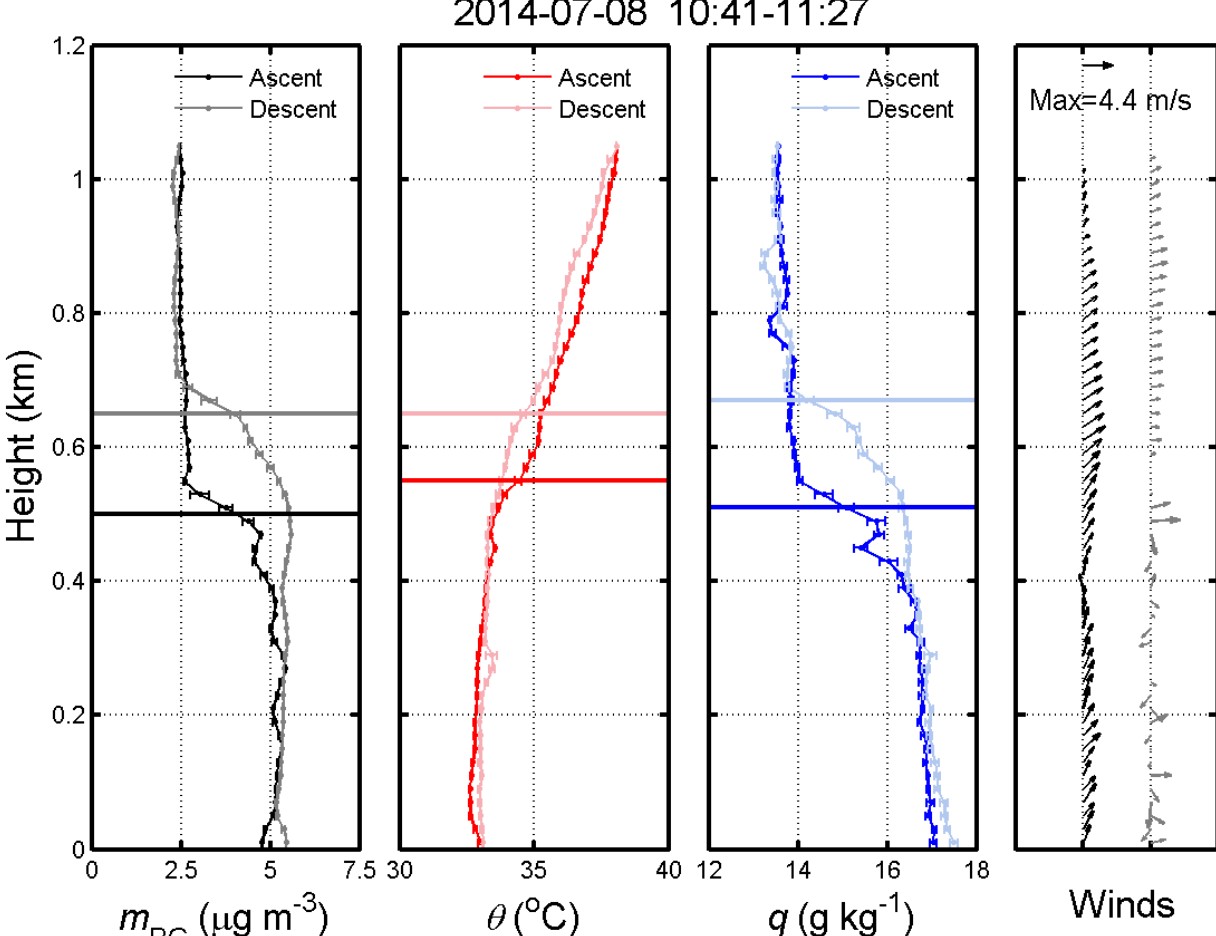

Figure 4. Vertical profiles of BC mass concentrations ($m_{BC}$), potential temperature ($\theta$), specific humidity ($q$) and winds on July 8, 2014 (10:41-11:27 LT). Profiles measured during the ascent are displayed in dark colors, while the descent in light colors. Dots indicate 20-m averaged data, with standard deviations in error bars. Heights of the mixing layer (ML) estimated from vertical profiles are given in horizontal lines. Arrows show vector mean horizontal winds, with northerly winds indicated by downward arrows. The wind vector for the maximum wind speed is marked with the text.

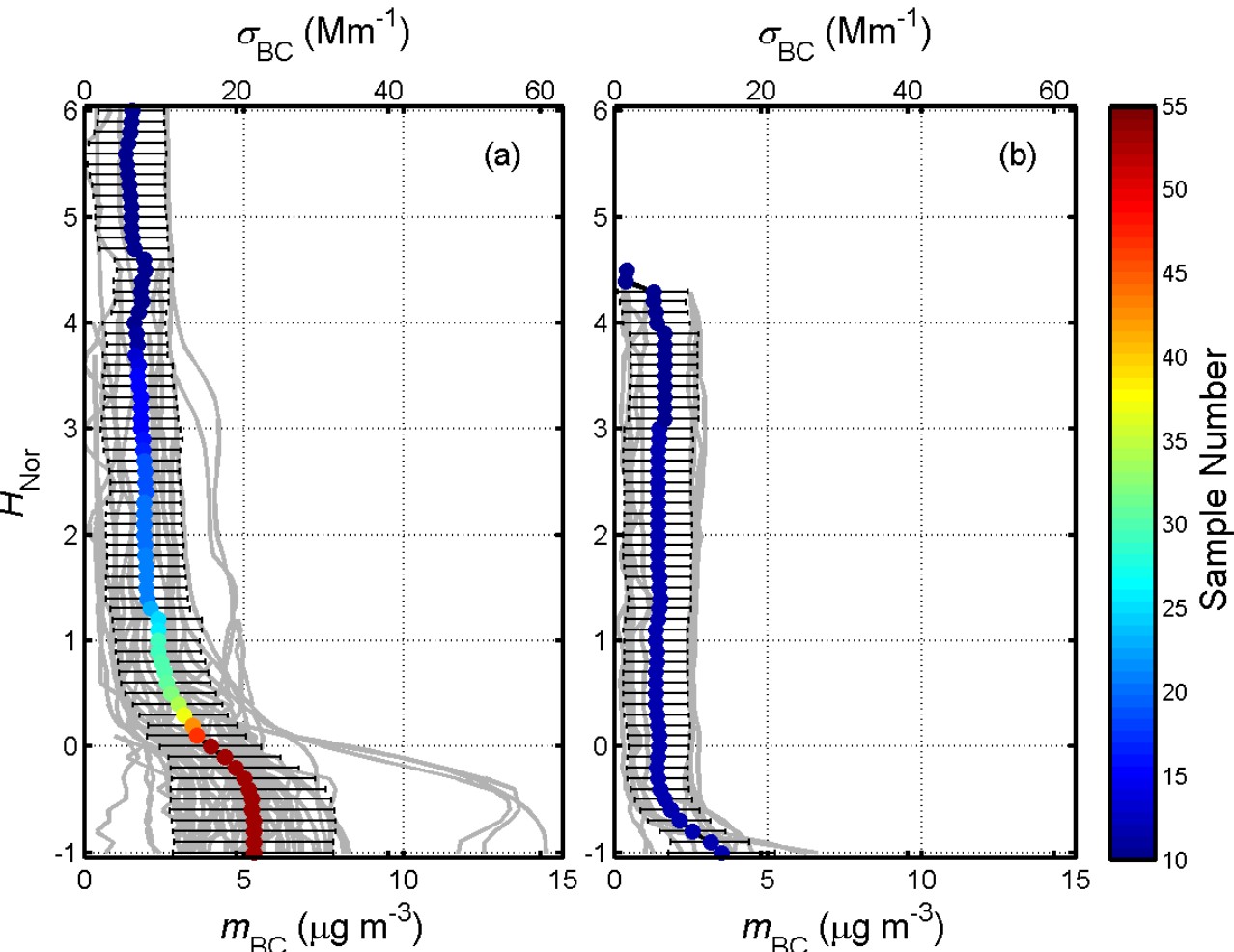

Figure 5. Vertical distributions of $m_{BC}$ (bottom axis) and $\sigma_{BC}$ (top axis) along the normalized height $H_{Nor}$ (a) in the daytime and (b) in the evening during the field campaign (grey lines). The average profile is shown in black line, with error bars to represent standard deviations. Average values are indicated by dots, with the color to show the number of samples at each layer.

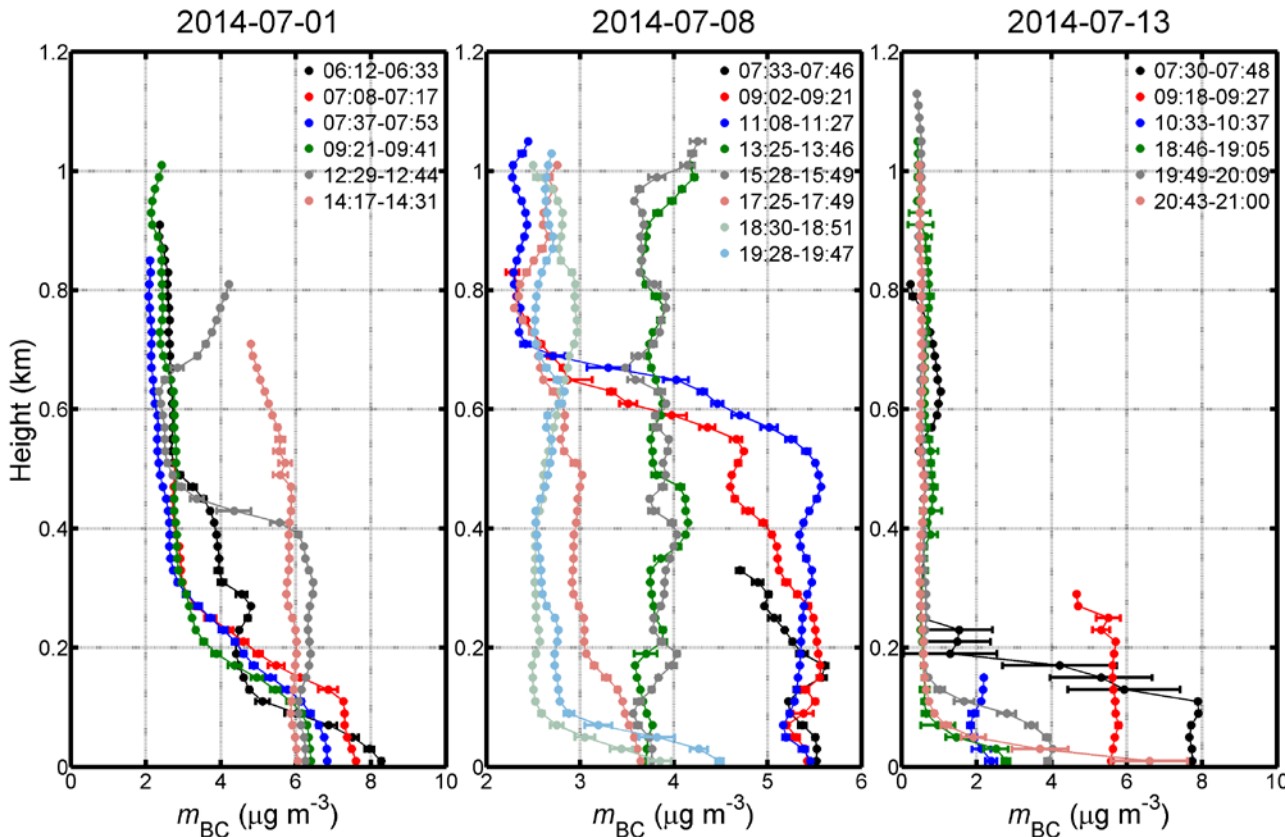

Figure 6. Vertical distributions of BC separately displayed for July 1, 8, and 13. Profiles collected at different time are shown in different colors. As in Fig. 4, dots represent 20-m averaged $m_{BC}$, with standard deviations given in error bars.

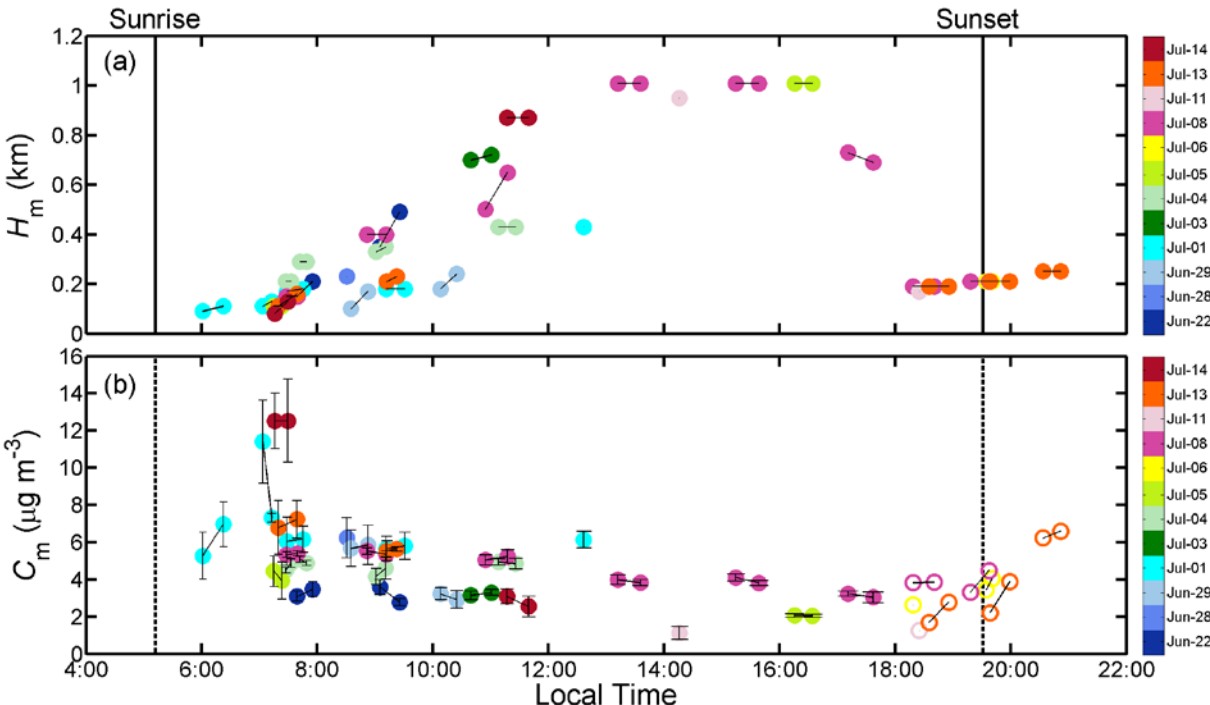

Figure 7. (a) $H_m$ identified from BC vertical profiles in the selected dataset. Measurements made on different days are marked by dots in different colors. $H_m$ from the ascent and descent of one launch are connected by black lines. (b) Average $m_{BC}$ within the ML ($C_m$) during the daytime (before 18:00 LT) are marked by solid dots. Average $m_{BC}$ within 20 m near the surface in the evening are marked by open dots.