# Peer review of "Vertical profiles of black carbon measured by a micro-aethalometer in summer in the North China Plain"

_Atmospheric Chemistry and Physics, 2016_

## Referee Comment (RC1) · Anonymous Referee #2 · 11 May 2016

This work reports vertical profiles of absorption during summer 2014 at a semirural area in the North China Plain. Profiles up to 1 km a.g.l. were measured by a single wavelength absorption monitor onboard a tethered balloon. Even though the area where the flights took place is reported as semi rural the measured concentrations are high and comparable to many polluted urban areas, as the authors state in the abstract (p1 l23-24). However in the manuscript no comparison is found between the surrounded areas and the area where the flights took place. Under such high concentrations, a validation in my point of view is required (see also my comment below).

In addition, the authors do not report the diurnal and, most important, temporal cycle of absorbing aerosol in that area. The reader is left questioned whether the reported concentrations correspond to the maximum, minimum or average concentrations in that area.

[Figure]

Even though the vertical profile analysis done in this work is, in my point of view, complete and comprehensible with assistive figures, it is not complemented by any discussion on the surrounding area and the particulate temporal variations. I also found disappointing that instead of showing a map of the measurement area, a reference is used instead. The manuscript must be complete by itself.

Additionally, the authors report a multiple scattering optical enhancement factor C equal to 2.52 using a mass attenuation cross-section equal to 12.5 m2 g-1. I strongly support enhancing that part, as it holds the greatest interest. The authors should give more details on how the C factor was calculated and provide, as a minimum, a graphical representation of the results. As an example, Ferrero et al. (2011) reports C equal to 2.05 instead. The AE-51 cannot operate on a 24/7 basis. It was never designed to do so. Therefore, the authors should provide more details on how they conducted the comparison.

On top of that, I recommend the authors to read Hyvarinen et al., (2013, doi:10.5194/amt-6-81-2013). In that work MAAP was reported to underestimate BC concentrations even when the sample was measured onto fresh sample-spots. Even though MAAP and AE51 use different methods to derive BC mass (or absorption) the sampling strategy remains the same; sample is accumulated on a sample spot. Even though the face velocity of AE51 is a factor of 3 lower than that of MAAP, it does not exclude that the effect described in Hyvarinen et al., (2013) does not take place here. The comparison performed here, but discussed too briefly, can shed a light on this. It will also add value to the vertical profiles shown in this work.

My last remark concerns the smoothing algorithm. I understand that on a single wavelength monitor a smoothing algorithm is primarily used to remove outliers and make measurements more presentable. Under this perspective the authors provide adequate information on the smoothing process. However, multi-wavelength miniature absorption instruments are on the way and a proper smoothing algorithm is essential in calculating the angstrom exponent, as an example. Therefore I strongly encourage the

authors to provide a comparison of unsmoothed dataset against those of the proposed algorithm and of Hagler et al. (2011).

Minor comments p1, L23-25. Please specify which polluted urban areas you are referring to.

p3, L5-10. Please add an image of the area, instead of a reference. Currently this work is somewhat incomplete.

p3, L19. what exactly does that statement means. Can you specify the conditions under which the AE-51 cannot operate. I am a bit surprised as this instrument has been used in drones moving with km h-1 speed.

p5, L9. please provide more details on how the artifacts were addressed. Start by briefly mentioning what artifacts you are referring to. This is not obvious to the reader. The manuscript should be complete.

p6, L5-6. how does this affect the measurements. What is the diurnal variation in this area?

p6, L29. I suspect a typo at Larzridis, 2011

On page 6 the authors mention two methods from estimating the boundary layer height. Please discuss the differences in PBL height from these two methods.

p7, L16-17. Were these frequent vertical profiles complemented by ground measurements somewhere close by?

p8, L1-2. Was there any measurements performed the previous night. What was the result?

---

## Referee Comment (RC2) · Anonymous Referee #1 · 13 May 2016

This paper report results of vertical profiles of black carbon aerosol collected in the North China Plain. The topic and the reported measurements are very important as vertical profile data of BC a globally scarse if compared with the high amount of ground-based observation. Thus the topic of this paper is of fundamental importance. It is suitable to be published on ACP after the authors raised the following points.

MAIN POINTS: Abstract (page 1 lines 12-20): the developement of the mixing layer is qualitatively described. Moreover it is reported that the mixing layer usually developed from 0.2 km up to 1 km (i.e. sunny days) and followed by a "collapse" during the evening. In a such situation a residual layer usually forms above the NBL making the concentration measured above the mixing layer not representative of a clean free troposphere. Please discuss also the possible importance of the residual layer formation on your measurements along the entire manuscript.

[Figure]

Section 2.2.1: the developed smoothing algorithm appears very promising. However, a deeper discussion here is called for. Especially it is necessary to compare the smoothing results with that can be obtained by the ONA (Hagler et al. (2011)) application. I strongly suggest to introduce a new picture to show the effect of the two data treatment on the raw collected BC data along vertical profiles. The reason for a such request comes from the fact that the Hagler at al. algorithm is based on the physical behaviour of the measured ATN in the Aethalometer, while the new smoothing algorithm reported in this paper appear only statistically based and somehow affected by the operator (i.e. "(6) Repeat step (1)-(5) for m times to obtain acceptable smoothed data"). Concerning the last point in brackets: have you defined a criteria for the "acceptable smoothed data"? How much is the threshold? How much is the loss in terms of vertical resolution of the data after the smoothing? I think the smoothing algorithm should be also discussed more quantitatively than did until now.

Section 2.2.2, page 5, line 8: "Details of the correction scheme developed for tackling with artifacts of AE-31 were described in Ran et al. (2016)". Note that Ran et al. (2016) is just a submitted paper. In the reference list the journal to which Ran et al. paper was submitted is missing. Please add it. Moreover, as the AE31 data could significantly change in function of the chosen correction function it is necessary to resume here at least the main points of the correction scheme adopted in Ran et al. as the paper is not yet available to the scientific community. With respect to this, depending on the chosen correction scheme (i.e. C factors for each wavelength of the AE31), the obtained angstrom exponent should change introducing an error on the retrieved $\sigma$MAAP,880nm. A quantitative assessment of the variability of $\sigma$MAAP,880nm depending on the chosen correction scheme for the AE31 is called for. Moreover, I strongly recommend an analysis of the error propagation of $\sigma$MAAP,880nm on the obtained C for the AE51. As a matter of fact the C factor of 2.52 is reported here without any statistical treatment of its uncertainty. Finally, no reference was made to the C value of 2.05 $\pm$ 0.03 for the AE51 reported in Ferrero et al. (2011a). It should very interesting to discuss the difference on the two C values in terms of the chemical composition of

the aerosol in the NCP with respect to the Europe.

Page 5, lines 12-13: "Measured $\sigma$AE-51,880nm (ATN<10) and calculated $\sigma$MAAP,880nm were linearly fitted with a correlation coefficient of 0.96 in a significant level (P<0.001), yielding a C value of 2.52": Please add the picture of this correlation.

Page 6, line 13, equation 6: "Hm was calculated from a sigmoid function that could well characterize typical daytime profile of mBC:". From this sentence it appears that Hm was calculated using equation 6. However, equation 6 requires as input both the mixing layer and the entrainment layer. This point is not clearly defined and needs to be specified. I also suggest to add a graphical example of the mixing layer determination using the sigmoid function. Finally a question: as you have both the potential temperature and wind profiles at disposal, have you ever thought to analyse the mixing layer also using the Richardson number approach?

MINOR POINTS: Page 7, lines 3-4: "the normalized height (HNor), which was calculated from h/Hm-1". In Ferrero et al. (2014) this analysis is explained. Add this reference at the end of the sentence.

Figure 2b: at Hnor=0 BC data are characterized by free troposphere concentration levels. I was a bit surprised about it. I expected that around Hnor=0 there was at least the end of the exponential deacrease of concentration starting from ground values. Could you comment it?

Page 3, lines 24-25 and equation 1: "to estimate aerosol absorption coefficients at the wavelength of 880 nm following"... Please note that $\sigma$AE-51,880nm is the attenuation coefficient and not the absorption coefficient as reported in many papers (i.e. starting from Weingartner et al. (2003)). Please correct the paper for this point.

---

## Author Response (AR1)

Dear Editor,

We greatly appreciate the editor's careful handling of the manuscript and the referees' valuable comments. We have addressed all comments point by point. Corresponding revisions have been made in the manuscript. A marked-up manuscript version has been provided after response to referees.

Response to referee #1:

We highly appreciate the referee's valuable comments and instructive suggestions. We have addressed each comment as below. Corresponding revisions have been made in the manuscript.

**This work reports vertical profiles of absorption during summer 2014 at a semirural area in the North China Plain. Profiles up to 1 km a.g.l. were measured by a single wavelength absorption monitor onboard a tethered balloon. Even though the area where the flights took place is reported as semi rural the measured concentrations are high and comparable to many polluted urban areas, as the authors state in the abstract (p1 l23-24). However in the manuscript no comparison is found between the surrounded areas and the area where the flights took place. Under such high concentrations, a validation in my point of view is required (see also my comment below). In addition, the authors do not report the diurnal and, most important, temporal cycle of absorbing aerosol in that area. The reader is left questioned whether the reported concentrations correspond to the maximum, minimum or average concentrations in that area.**

**Even though the vertical profile analysis done in this work is, in my point of view, complete and comprehensible with assistive figures, it is not complemented by any discussion on the surrounding area and the particulate temporal variations. I also found disappointing that instead of showing a map of the measurement area, a reference is used instead. The manuscript must be complete by itself.**

We thank the referee for the helpful comments. The figure mentioned in the manuscript shows the spatial distribution of average aerosol optical depth (AOD) at 550 nm from the level 2 version of the Moderate Resolution Imaging Spectroradiometer (MODIS) data (Levy and Hsu, 2015) during April 2013 and March 2015 in the NCP (Fig.1 in Ran et al., 2016). Considering that no extra information was given, we did not put this figure in the manuscript. Instead, a description of the sampling site (38°14'N, 115°44'E, miswritten as 38.14°N, 115.44°E there) was present with the cited reference Ran et al. (2016), which has just been published in Atmospheric Environment. However, we agree with the referee that the manuscript should be complete by itself. Thus, an illustration of the AOD distributions in different months of 2014 in the NCP is given in the revised manuscript to provide a direct view of the site and its surroundings. It could also cast some light on the seasonal variation of ambient aerosols in the area where launches of the tethered balloon were carried out, since measurements of absorbing aerosols are unavailable to obtain the temporal variation in that area.

The feature of regional aerosol pollution with clearly defined pollution centers could be recognized from the AOD distributions in different months of the year 2014 (Fig. R1, also as Fig. 1 in the revised manuscript). The level of AOD in Raoyang could well represent regional aerosol conditions in central NCP in July 2014, the month when most of tethered balloon flights took place (Fig. R1c). Moreover, given severe regional aerosol pollution in the NCP, it was understandable that reported BC concentrations in the semirural site of Raoyang were quite high and even comparable to some polluted urban centers in the world.

However, it should be noted that the seasonal cycle of ambient aerosols depends on both the aerosol loading and the relative humidity. As a consequence, the seasonal variation of ambient aerosols obtained from the AOD distributions might not be able to fully represent that of the aerosol loading, not even to mention that of absorbing aerosols. Besides, the diurnal cycle of absorbing aerosols would also be affected by the evolution of the mixing layer and meteorological parameters such as wind speed and direction. Therefore, it is complicated to draw a definite conclusion only from the AOD dataset about to what extent reported BC concentrations represent the spatiotemporal variability in this area.

To gain more confidence in the spatial representativeness of the sampling site, BC emissions (0.25°×0.25°) from four sectors in 2012 in the NCP region were obtained from the multi-resolution emission inventory for China (MEIC, http://www.meicmodel.org) model. Figure R2 (also as Fig. S1 in the supplement) shows the spatial distribution of BC emissions from the sector of industry, power, residential activity and transportation. Centers of intense emissions could be clearly seen, generally coincided with pollution centers in the AOD map. According to the emission inventory of BC, the site of Raoyang could represent the regional situation, also might be influenced by several centers with heavy industry and transportation emissions in the surrounding area.

[Figure]

Figure R1. The spatial distribution of averaged MODIS aerosol optical depth (AOD) at 550 nm in (a) January; (b) April; (c) July; (d) October, 2014 in the NCP. The locations of the semirural site Raoyang and major cities are respectively marked by square and dots. Only grids with the fraction of valid data exceeding 30% in the month are shown.

[Figure]

Figure R2. The spatial distribution of BC emissions from the sector of (a) Industry; (b) Power; (c) Residential Activity; (d) Transportation, based upon the emission inventory from the MEIC Model.

Furthermore, seasonal and diurnal variations of surface BC mass concentrations were analyzed on basis of about six-year measurements (from 2006 January to 2012 July with the data completeness of 77.5%) using an aethalometer (Model AE-31, Magee Scientific, USA) with a temporal resolution of 5 min in Gucheng, about 90 km north of Raoyang (Zhang et al., 2015). BC mass concentrations averaged about $9.6\pm8.4$ μg m$^{-3}$ during 2006 and 2012 in Gucheng, with a lower level in summer and spring. The diurnal cycle of BC mass concentrations indicated higher values at night and a low valley in the afternoon. Similar to Raoyang, Gucheng is a semirural site surrounded by city clusters in the NCP (Fig. R1). Measurements at both sites showed a high level of BC that was comparable to many polluted urban areas, which might be surprising but understandable in such a severely polluted region. Also due to the similarity in the AOD level and BC emissions at both sites, BC mass concentrations measured in summer at Raoyang might probably be lower than that in other seasons.

Following above discussions, we have revised the manuscript as:

**P3, L11,** "The spatial distribution of average aerosol optical depth (AOD) at 550 nm acquired from the level 2 version of the Moderate Resolution Imaging Spectroradiometer (MODIS) data (Levy and Hsu, 2015) is also displayed in Fig. 1. The feature of severe regional aerosol pollution with clearly defined pollution centers could be recognized from the AOD distributions in different months of the year 2014. The level of AOD in Raoyang could well represent regional aerosol conditions in NCP in July 2014, the month when most of tethered balloon flights took place (Fig. 1c). The AOD distributions could also cast some light on the seasonal variation of ambient aerosols in the area where launches of the tethered balloon were carried out, since measurements of absorbing aerosols are unavailable to obtain the temporal variation in that area. However, it should be noted that the seasonal cycle of ambient aerosols depends on both the aerosol loading and the relative humidity. As a consequence, it is complicated to draw a definite conclusion only from the AOD dataset about to what extent reported BC concentrations in this study represent the spatiotemporal variability in the area. A further examination on BC emissions (0.25°×0.25°) from four sectors (industry, power, residential activity, and transportation) in 2012 in the NCP region was performed on basis of emission data generated from the multi-resolution emission inventory for China (MEIC, http://www.meicmodel.org) model. It could also tell from the emission inventory of BC, that aerosol conditions at the site of Raoyang well represents the regional situation, with influences from several nearby emission centers (Fig. S1).

Moreover, it was found that another semirural site Gucheng, about 90 km north of Raoyang, shared a similarity with Raoyang in the AOD level as well as BC emissions. Seasonal and diurnal variations of surface BC mass concentrations in Gucheng were analyzed on basis of about six-year measurements (from 2006 January to 2012 July with the data completeness of 77.5%) using an aethalometer (Model AE-31, Magee Scientific, USA) with a temporal resolution of 5 min (Zhang et al., 2015). BC mass concentrations averaged about $9.6\pm8.4$ μg m$^{-3}$ during 2006 and 2012 in Gucheng, with a lower level in summer and spring. The diurnal cycle of BC mass concentrations indicated higher values at night and a low valley in the afternoon. Thus, it might be expected that BC mass concentrations measured in summer at Raoyang were probably lower than that in other seasons."

**Additionally, the authors report a multiple scattering optical enhancement factor C equal to 2.52 using a mass attenuation cross-section equal to 12.5 m$^2$ g$^{-1}$. I strongly support enhancing that part, as it holds the greatest interest. The authors should give more details on how the C factor was calculated and provide, as a minimum, a graphical representation of the results. As an example, Ferrero et al. (2011) reports C equal to 2.05 instead. The AE-51 cannot operate on a 24/7 basis. It was never designed to do so. Therefore, the authors should provide more details on how they conducted the comparison.**

We totally agree with the referee that we should strengthen the part regarding the calculation of the *C* factor in the manuscript, considering its importance to data processing and subsequent discussions. We also agree with the referee that AE-51 is in principle not designed to be operated around the clock. Continuous comparative measurements using AE-51, AE-31, and MAAP were however carried out, in order to obtain a proper size of the

data for the calculation of the *C* factor in limited time period. A daily check of the flow rate was performed using a Gilibrator-2 Diagnostic Kit (Sensidyne, USA), to ensure the stability of the flow and thus data quality.

The calculation of the *C* factor was based upon concomitant aerosol absorption measurements using AE-51, AE-31 and MAAP for about 1 week in Beijing. The actual wavelength of MAAP is 637 nm instead of the nominal wavelength of 670 nm (Müller et al., 2011), as pointed out by one of the referees for Ran et al. (2016). Accordingly, we have corrected all related results in the revised manuscript.

Three steps were taken to obtain the *C* factor. Firstly, aerosol absorption Angström exponent ($\alpha$) for the spectrum over the span of 637 and 880 nm was estimated using AE-31 measurements at the channels of 660 and 880 nm. Attenuation coefficients at 660 and 880 nm were corrected to derive absorption coefficients $\sigma_{AE-31,660nm}$ and $\sigma_{AE-31,880nm}$. The correction scheme was the same as that in Ran et al. (2016). Then, $\alpha$ was calculated from:

$$\alpha = \frac{\ln(\sigma_{AE-31,660nm}) - \ln(\sigma_{AE-31,880nm})}{\ln(880) - \ln(660)}. \tag{R1}$$

Secondly, $\sigma_{MAAP,880nm}$ was quantified from $\sigma_{MAAP,637nm}$ following the spectral dependence of aerosol absorption coefficients in the form of $\lambda^{-\alpha}$:

$$\sigma_{MAAP,880nm} = \sigma_{MAAP,637nm} \times (\frac{880}{637})^{-\alpha}. \tag{R2}$$

Finally, $\sigma_{MAAP,880nm}$ were taken as real values of absorption coefficients at 880 nm. Reduced major axis regression of attenuation coefficients $\sigma_{AE-51,880nm}$ (ATN<10) measured by AE-51 and absorption coefficients $\sigma_{MAAP,880nm}$ calculated from MAAP and AE-31 yielded the *C* factor of 2.98±0.05 with 95% confidence (Fig. R3, also as Fig. 2 in the revised manuscript).

[Figure]

Figure R3. Reduced major axis regression of attenuation coefficients $\sigma_{AE-51,880nm}$ (ATN<10) measured by AE-51 and absorption coefficients $\sigma_{MAAP,880nm}$ calculated from concomitant MAAP and AE-31 measurements in the comparative test.

Accordingly, revisions have been made in the manuscript as:

**P6, L6,** "The *C* factor was derived from a surface comparative test for about 1 week in Beijing among AE-51, a 7-wavelength aethalometer (Model AE-31, Magee Scientific, USA) and a multi-angle absorption photometer (MAAP, Model 5012, Thermo, USA). Continuous operation of AE-51 was carried out in order to obtain a proper size of the dataset in limited time period, despite that AE-51 is in principle not designed to be operated around the clock. A daily check of the flow rate was performed using a Gilibrator-2 Diagnostic Kit (Sensidyne, USA) to ensure the stability of the flow and thus data quality."

**P6, L30,** "Three steps were taken to obtain the *C* factor. Firstly, aerosol absorption Angström exponent ($\alpha$) over the spectrum span of 660 and 880 nm was derived from absorption coefficients $\sigma_{\text{AE-31,660nm}}$ and $\sigma_{\text{AE-31,880nm}}$, which were corrected from attenuation coefficients at 660 and 880 nm measured by AE-31. Hence, $\alpha$ was calculated from:

$$\alpha = \frac{\ln(\sigma_{\text{AE-31,660nm}}) - \ln(\sigma_{\text{AE-31,880nm}})}{\ln(880) - \ln(660)} . \tag{2}$$

Secondly, $\alpha$ for the spectrum of 660 and 880 nm was used to represent $\alpha$ over the span of 637 and 880 nm. Therefore, $\sigma_{\text{MAAP,880nm}}$ was quantified from measured $\sigma_{\text{MAAP,637nm}}$ following the spectral dependence of aerosol absorption coefficients in the form of $\lambda^{-\alpha}$:

$$\sigma_{\text{MAAP,880nm}} = \sigma_{\text{MAAP,637nm}} \times (\frac{880}{637})^{-\alpha} , \tag{3}$$

Finally, reduced major axis regression of attenuation coefficients $\sigma_{\text{AE-51,880nm}}$ (ATN<10) measured by AE-51 and absorption coefficients $\sigma_{\text{MAAP,880nm}}$ calculated from MAAP and AE-31 yielded the *C* factor of 2.98±0.05 with 95% confidence (Fig. 2)."

**On top of that, I recommend the authors to read Hyvarinen et al., (2013, doi:10.5194/amt-6-81-2013). In that work MAAP was reported to underestimate BC concentrations even when the sample was measured onto fresh sample-spots. Even though MAAP and AE51 use different methods to derive BC mass (or absorption) the sampling strategy remains the same; sample is accumulated on a sample spot. Even though the face velocity of AE51 is a factor of 3 lower than that of MAAP, it does not exclude that the effect described in Hyvarinen et al., (2013) does not take place here. The comparison performed here, but discussed too briefly, can shed a light on this. It will also add value to the vertical profiles shown in this work.**

We appreciate the referee's valuable suggestion. Hyvärinen et al. (2013) reported an artifact in measuring BC mass concentrations using MAAP, namely, an underestimation after a spot change. This artifact was considered possibly to be associated with erroneous dark counts in the transmitted light photodetector and an instrument internal averaging procedure of the photodetector raw signals. It was stated that this artifact could be observed with a BC mass accumulation rate, as the product of BC mass concentrations and the flow rate of MAAP, larger than 0.04 µg min$^{-1}$, which corresponds to 3 µg m$^{-3}$ at the flow rate of 1 m$^3$ h$^{-1}$. However in this study, no apparent underestimation of BC mass concentrations was found at the beginning of a spot, in comparison with the last several samples collected on the previous spot, even for cases where BC mass concentrations exceeded 8 ug m$^{-3}$, which stood for an accumulation rate of 0.107 µg min$^{-1}$ with the flow rate of 0.8 m$^3$ h$^{-1}$ here (Fig. R4, also as Fig. S2 in the supplement). Consequently, no corrections were needed for the MAAP measurements in this study as in Hyvärinen et al. (2013). As for aethalometers, either AE-31 or AE-51, no such effect has been encountered in current experiment and previous studies, also no related literature has been found. Nonetheless, we have added some discussions on this issue in the revised manuscript to leave it open to the community.

**P6, L23,** "Hyvärinen et al. (2013) reported an artifact of underestimated BC mass concentrations after a spot change and attributed it to yet unconfirmed causes as erroneous dark counts in the transmitted light photodetector and an instrument internal averaging procedure of the photodetector raw signals. It was stated that this artifact

could be observed with a BC mass accumulation rate, as the product of BC mass concentrations and the flow rate of MAAP, larger than 0.04 µg min$^{-1}$, which corresponds to 3 µg m$^{-3}$ at the flow rate of 1 m$^3$ h$^{-1}$. However, no apparent underestimation of BC mass concentrations was found in this study, even for cases where BC mass concentrations exceeded 8 ug m$^{-3}$ (Fig. S2). Consequently, measured $\sigma_{MAAP,637nm}$ without any corrections were used for subsequent calculations and taken as real values."

[Figure]

Figure R4. A comparison between measurements using MAAP across filter spot changes for cases where BC mass concentrations exceeded 8 ug m$^{-3}$. Data points (with a temporal resolution of 1 min) collected before and after a spot change were denoted by the same marker in the same color.

**My last remark concerns the smoothing algorithm. I understand that on a single wavelength monitor a smoothing algorithm is primarily used to remove outliers and make measurements more presentable. Under this perspective the authors provide adequate information on the smoothing process. However, multi-wavelength miniature absorption instruments are on the way and a proper smoothing algorithm is essential in calculating the angstrom exponent, as an example. Therefore I strongly encourage the authors to provide a comparison of unsmoothed dataset against those of the proposed algorithm and of Hagler et al. (2011).**

We thank the referee for this helpful comment. We proposed in this study a new smoothing algorithm, Fluctuation Minimizing Smoothing (FMS). With properly chosen smoothing window and smoothing count, data fluctuation resulted from the high temporal resolution could be effectively minimized. A comparison was made between unsmoothed data, smoothed data using the FMS approach in this study and the ONA method in Hagler et al. (2011), as well as 20-m averaged data using those two algorithms. Two examples are given below to show that both algorithms could well deal with noises in the signal without introducing artificial features (Fig. R5, also as Fig. 3 in the revised manuscript). However, the FMS method was found to be more capable of reliably removing outliers in some cases (e.g., Fig. R5d-R5f). The comparison indicated that the FMS procedure reduced more fluctuation, meanwhile still preserved reasonable variability of the data.

[Figure]

Figure R5. (a) Unsmoothed BC mass concentrations measured with a temporal resolution of 1 s on July 1, 2014 (09:02-09:41 LT). Data points collected from the ascending and descending process are respectively marked in black and grey dots. (b) Smoothed BC mass concentrations using two algorithms. Data points processed by the ONA method are displayed in large pink dots for the ascent and in light green color for the descent. Data points processed by the FMS method are denoted by small red dots for the ascent and green dots for the descent. (c) 20-m averaged profiles based upon smoothed data using two algorithms. Dots indicate 20-m averages, with standard deviations in error bars. Results from the ONA and FMS methods are respectively given in the color of light blue and blue for the ascent, while in the color of light purple and purple for the descent. (d)-(f) Measured and processed BC vertical profiles on July 8, 2014 (08:41-09:21 LT). The caption is the same as that in (a)-(c).

Following the referee's suggestion, we have revised the manuscript as:

**P5, L23,** "A comparison was made between unsmoothed data, smoothed data using the FMS approach in this study and the ONA method in Hagler et al. (2011), as well as 20-m averaged data using those two algorithms. It was found that both algorithms could properly deal with data fluctuation caused by instrumental noises without introducing artificial features (e.g., Fig. 3a-3c). However, the FMS method seemed to be more capable of reliably removing outliers in some cases (e.g., Fig. 3d-3f). The comparison indicated that the FMS procedure could effectively reduce data fluctuation while still preserve reasonable variability of the profile."

Moreover, we have corrected a mistake in the description of the smoothing algorithm.

**P5, L8,** "…, where $i=1,2,…,N-1,N$ and $j=1,2,…,n-1,n$;"

**Minor comments p1, L23-25. Please specify which polluted urban areas you are referring to.**

We have specified the polluted areas in the revised manuscript as:

**P1, L22,** "During the field campaign, $C_m$ averaged about 5.16±2.49 µg m$^{-3}$, with a range of 1.12 to 14.49 µg m$^{-3}$, comparable with observational results in many polluted urban areas such as Milan in Italy and Shanghai in China."

**p3, L5-10. Please add an image of the area, instead of a reference. Currently this work is somewhat incomplete.**

We have added a figure as Fig. 1 in the revised manuscript to show the site and its surrounding area, as given in the response to the first comment.

**p3, L19. what exactly does that statement means. Can you specify the conditions under which the AE-51 cannot operate. I am a bit surprised as this instrument has been used in drones moving with km h$^{-1}$ speed.**

We thank the referee for pointing out the ambiguous statement. Actually, the reason for lack of data is not that AE-51 itself could not operate under strong winds, but that the violent swing of the tethered balloon and the instrument caused poor data quality. As long as the tethered balloon could maintain stable and the instrument kept its posture, valid data could be obtained even under continuous winds if not so strong. However, the randomly varying wind gusts could bring about large artificial variations in measured data. During one launch, the drier prior to the inlet of AE-51 was even torn apart by strong winds. Under such situations, the measurements were considered invalid and the data were discarded. To be clearer, we have revised the manuscript as:

**P4, L11,** "Lack of data for several ascending or descending processes was primarily caused by discarding invalid data under wind gusts, which led to the violent swing of the tethered balloon and poor data quality."

**p5, L9. please provide more details on how the artifacts were addressed. Start by briefly mentioning what artifacts you are referring to. This is not obvious to the reader. The manuscript should be complete.**

We have given a more detailed description of the artifacts and the way they were addressed in the revised manuscript as:

**P5, L30,** "Measured $\sigma_{AE-51,880nm}$ suffered from systematic biases introduced by the filter-based technique. In order to determine BC absorption coefficients ($\sigma_{BC}$) from $\sigma_{AE-51,880nm}$, corrections were required to tackle with three types of artifacts. The shadowing effect, an artifact that results in gradual artificial reduction in $\sigma_{AE-51,880nm}$ due to the saturation of the filter with increasing aerosol loading, leads to an underestimation of $\sigma_{BC}$ and a discontinuity after changing to a new sample spot (Weingartner et al., 2003). Various methods have been developed to address the shadowing effect (Weingartner et al., 2003; Virkkula et al., 2007; Ran et al., 2016). However, this artifact could be neglected in this study, since no ATN exceeded 20 with a new filter for each launch. The other two artifacts cause an overestimation of $\sigma_{BC}$ by enhancing light attenuation, either due to aerosol scattering or the multiple scattering of filter fibers (Weingartner et al., 2003; Arnott et al., 2005; Schmid et al., 2006; Collaud Coen et al., 2010). A correction factor ($C$) was needed to correct these two artifacts."

**p6, L5-6. how does this affect the measurements. What is the diurnal variation in this area?**

The dataset was largely collected in the morning and evening as could be found in Table 1, since a stable condition suitable for launches of the tethered balloon were often encountered then. Around the midday and in the afternoon, surface wind speed usually increased (Fig. R6). It often became very difficult to launch the tethered balloon, due to strong winds and sometimes strong vertical wind shear even not high above the ground.

[Figure]

Figure R6. The average diurnal cycle of surface wind speed measured by the automatic weather station at the site on days when tethered balloon flights took place (black line). Color shows the probability of wind speed with an interval of 0.5 m s$^{-1}$ encountered at different time with an interval of 5 min.

**p6, L29. I suspect a typo at Larzridis, 2011**

Thank you. We have corrected the typo in the revised manuscript.

**On page 6 the authors mention two methods from estimating the boundary layer height. Please discuss the differences in PBL height from these two methods.**

Following the referee's suggestion, a comparison was made between the mixing height determined from profiles of $m_{BC}$ using the gradient method ($H_{m,BC,gradient}$) and the sigmoid function ($H_{m,BC,sigmoid}$) for typical daytime BC vertical profiles. It was found that results from the two methods generally agreed quite well with each other, with a difference of less than 2 % (Fig. R7, also as Fig. S5 in the supplement). In addition to reliably estimating the mixing height as the gradient method, the sigmoid function could also directly provide parameters including $C_{ms}$, $C_{fs}$, and $H_e$. Therefore, the sigmoid function was applied in the manuscript to obtain all those parameters for typical daytime BC vertical profiles.

Above discussions have been added in the revised manuscript. We have also added Fig. S3 in the supplement (Fig. R8 here) to show the demonstrated well agreement among mixing heights estimated from vertical profiles of $m_{BC}$ using the gradient method ($H_{m,BC,gradient}$) and that of $\theta$ ($H_{m,\theta}$) and $q$ ($H_{m,q}$) for the entire dataset.

**P8, L21,** "The mixing height could be determined by applying the gradient method to the entire dataset (Seibert et al., 2000; Kim, et al., 2007). Generally, the mixing height determined from profiles of $m_{BC}$ ($H_{m,BC,gradient}$) agreed well with that from profiles of $\theta$ ($H_{m,\theta}$) and $q$ ($H_{m,q}$) as shown in Fig. S3."

**P8, L30,** "A comparison was made between $H_{m,BC,gradient}$ and $H_{m,BC,sigmoid}$ for typical daytime BC vertical profiles. Results from the two methods agreed quite well with each other, with a difference of less than 2 % (Fig. S5). In addition to reliably estimating the mixing height as the gradient method, the sigmoid function could also directly determine parameters including $C_{ms}$, $C_{fs}$, and $H_e$. Therefore, the sigmoid function was chosen to obtain all parameters for typical daytime BC profiles."

[Figure]

Figure R7. A comparison between mixing heights estimated from vertical profiles of $m_{BC}$ using the gradient method ($H_{m,BC,gradient}$) and the sigmoid function ($H_{m,BC,sigmoid}$) for typical daytime profiles.

[Figure]

Figure R8. A comparison among mixing heights estimated from vertical profiles of $m_{BC}$ using the gradient method ($H_{m,BC,gradient}$) and that of $\theta$ ($H_{m,\theta}$) and $q$ ($H_{m,q}$) for the entire dataset.

**p7, L16-17. Were these frequent vertical profiles complemented by ground measurements somewhere close by?**

Unfortunately, ground measurements of black carbon mass concentrations are not available.

**p8, L1-2. Was there any measurements performed the previous night. What was the result?**

There were no measurements carried out the previous night of July 1 due to strong winds. The experiment was designed to collect nighttime vertical profiles, and also to conduct continuous measurements across several days, but the outcome was eventually decided by the weather.

**References**

[revised manuscript text omitted]

Response to referee #2:

We highly appreciate the referee's valuable comments and instructive suggestions. We have addressed each comment as below and corresponding revisions have been made in the manuscript.

**This paper report results of vertical profiles of black carbon aerosol collected in the North China Plain. The topic and the reported measurements are very important as vertical profile data of BC a globally scarce if compared with the high amount of ground-based observation. Thus the topic of this paper is of fundamental importance. It is suitable to be published on ACP after the authors raised the following points.**

**MAIN POINTS: Abstract (page 1 lines 12-20): the development of the mixing layer is qualitatively described. Moreover it is reported that the mixing layer usually developed from 0.2 km up to 1 km (i.e. sunny days) and followed by a "collapse" during the evening. In a such situation a residual layer usually forms above the NBL making the concentration measured above the mixing layer not representative of a clean free troposphere. Please discuss also the possible importance of the residual layer formation on your measurements along the entire manuscript.**

We thank the referee for the valuable suggestion. We agree with the referee that the existence of a residual layer would make measured BC mass concentrations ($m_{BC}$) above the mixing layer not representative of a clean free troposphere. As stated in the manuscript, average $m_{BC}$ in free troposphere could reach 2~3 μg m$^{-3}$ under polluted conditions, otherwise usually well below 1 μg m$^{-3}$ under clean conditions. The case study of vertical profiles measured on July 1 (Fig. 6) showed a polluted layer with a thickness of 0.3 km in the morning, possibly a residual layer formed the day before. The level of $m_{BC}$ above the polluted layer was also as high as ~2 μg m$^{-3}$. The case study of vertical profiles measured on July 8 (also in Fig. 6) showed how vertical profiles of $m_{BC}$ evolved with the development of the planetary boundary layer (PBL). A relatively high level of $m_{BC}$ was found above the NBL where the remnant of the daytime mixing layer after its collapse might be traced. However, analysis regarding the impact of the residual layer formed in the previous evening on measurements on the next day is difficult to carry out without continuous measurements from the previous day. Also the characteristics of the residual layer should be affected by the advection. The role of the residual layer in affecting the evolution of the PBL still stays controversial, though it has been consented that BC could heat the PBL and intensify atmospheric stability. Ding et al. (2016) demonstrated the importance of the "dome effect" of BC in the PBL especially the upper PBL, suppressing the PBL height and enhancing haze pollution within a lower PBL. However, Zhang et al. (2012) indicated a limited warming effect of BC in an elevated aerosol layer, and also limited induced increase in the strength of atmospheric inversion.

Corresponding discussions in the revised manuscript include:

**P10, L8,** "This might imply the existence of a polluted residual layer above the stable surface layer formed after the sunset in previous evening, yet unable to be further discussed without continuous measurements from the day before. Also the characteristics of $m_{BC}$ in FT should be affected by the advection."

**P10, L17,** "Sometimes, a residual layer with a relatively high level of $m_{BC}$ (>2 μg m$^{-3}$) could be formed above the NBL where the remnant of the daytime mixing layer might be traced after its collapse (e.g., profiles on July 8). This would undoubtedly have an impact on measured $m_{BC}$ above the mixing layer on the next day, leading to a polluted background in FT (e.g., in the morning of July 1 and 8). The role of the residual layer in affecting the evolution of the PBL still stays controversial, though it has been consented that BC could heat the PBL and intensify atmospheric stability. Ding et al. (2016) demonstrated the importance of the "dome effect" of BC in the PBL especially the upper PBL, suppressing the PBL height and enhancing haze pollution within a lower PBL. Whereas in Zhang et al. (2012), a limited warming effect of BC in an elevated aerosol layer and limited induced increase in the strength of atmospheric inversion were indicated."

**Section 2.2.1: the developed smoothing algorithm appears very promising. However, a deeper discussion here is called for. Especially it is necessary to compare the smoothing results with that can be obtained by the ONA (Hagler et al. (2011)) application. I strongly suggest to introduce a new picture to show the effect**

**of the two data treatment on the raw collected BC data along vertical profiles. The reason for a such request comes from the fact that the Hagler at al. algorithm is based on the physical behavior of the measured ATN in the Aethalometer, while the new smoothing algorithm reported in this paper appear only statistically based and somehow affected by the operator (i.e. "(6) Repeat step (1)-(5) for m times to obtain acceptable smoothed data"). Concerning the last point in brackets: have you defined a criteria for the "acceptable smoothed data"? How much is the threshold? How much is the loss in terms of vertical resolution of the data after the smoothing? I think the smoothing algorithm should be also discussed more quantitatively than did until now.**

We agree with the referee that a comparison between processed data using the two smoothing algorithms, Fluctuation Minimizing Smoothing (FMS) method proposed in this study and the ONA method in Hagler et al. (2011), should be made to show the similarity and differences in effects of the two approaches on unsmoothed data. As found in the two cases displayed in Fig. R1 (also as Fig. 3 in the revised manuscript), generally, both algorithms properly treated data fluctuation and largely improved the presented data. However, the FMS procedure seemed to be more capable of reliably reducing data fluctuation without losing details on the variability of vertical profiles (Fig. R1d-R1f).

[Figure]

Figure R1. (a) Unsmoothed BC mass concentrations measured with a temporal resolution of 1 s on July 1, 2014 (09:02-09:41 LT). Data points collected from the ascending and descending process are respectively marked in black and grey dots. (b) Smoothed BC mass concentrations using two algorithms. Data points processed by the ONA method are displayed in large pink dots for the ascent and in light green color for the descent. Data points processed by the FMS method are denoted by small red dots for the ascent and green dots for the descent. (c) 20-m averaged profiles based upon smoothed data using two algorithms. Dots indicate 20-m averages, with standard deviations in error bars. Results from the ONA and FMS methods are respectively given in the color of light blue and blue for the ascent, while in the color of light purple and purple for the descent. (d)-(f) Measured and processed BC vertical profiles on July 8, 2014 (08:41-09:21 LT). The caption is the same as that in (a)-(c).

The FMS method was devised to smooth the highly temporally resolved data (1 s) from AE-51. Similar to the ONA method (Hagler et al., 2011), the FMS approach is also principally based upon the physical behavior of measured ATN. Usually, ATN is supposed to increase with time. However, reported ATN might largely fluctuate due to limited sampling on the filter and instrumental noises such as that from the light source, the detector, electronics, the flow rate and unstable posture. Despite that BC values determined from fluctuated ATN might drastically vary, large positive/negative BC pairs would always be found and counterbalance each other within a few seconds. Therefore, the FMS method minimizes data fluctuation by finding pairs of BC values that differ largely with each other within a few seconds and making a compromise of them. The smoothing window $n$ used to search for pairs and the smoothing count used to repeat the smoothing were empirically chosen. Normally, data fluctuation is already compensated within 5 s, according to what has been observed in data processing.

To address the loss of the vertical resolution of processed data using the FMS method, the contribution from neighboring data points to the weighted average of each target point was calculated. In the FMS procedure, each data point was averaged within a range of $2n$ data points, where $n$ is the smoothing window. The average process was repeated by $m$ times, where $m$ is the smoothing count. With $n$ to be 5 and $m$ to be 1 or 5, average weight function for each profile was calculated and the result was similar among profiles. When $m$ was set to be 1, the average of the target point was mostly contributed from neighboring data points within about 11 seconds, according to a weight of 80%. This consequently led to a vertical resolution of about 22 m for the ascent and 11 m for the descent after smoothing. Similarly, the vertical resolution was about 50~60 m for the ascent and 25~30 m for the descent when $m$ was set to be 5. Figure R2 presents average weight function for two cases as given in Fig. R1. Different choices of the smoothing count gave a similar pattern of vertical profiles, but with some differences in details. To achieve a better smoothing for further calculations, the smoothing count was chosen to be 5 in this work.

[Figure]

Figure R2 Average weight function of neighboring data points to show their contribution to the weighted average of each target point for the case on July 1, 2014 (09:02-09:41 LT) with the smoothing window $n$ to be 5, (a) the smoothing count $m$ was set to be 1; (b) $m$ was set to be 5. Similar to the first case, average weight function for the case on July 8, 2014 (08:41-09:21 LT), (c) the smoothing count $m$ was set to be 1; (d) $m$ was set to be 5.

According to above discussions, we have revised the manuscript as:

**P4, L30,** "In this study, data dispersion due to high temporal resolution was treated by a new smoothing algorithm, Fluctuation Minimizing Smoothing (FMS). Similar to the ONA method, the FMS approach is also principally based upon the physical behavior of measured ATN. Despite that BC values determined from fluctuated ATN might drastically vary, large positive/negative pairs of BC values would always be found and counterbalance each other within a few seconds. Therefore, the FMS approach was devised to find pairs of BC values that differ largely with each other within a few seconds and make a compromise."

**P5, L15,** "The smoothing window $n$ and the smoothing count $m$ were empirically chosen during the calculation. It should be kept in mind that using improper large $n$ or $m$ might wipe off some natural variations, although it will always give a smoother result. $n$ should be set to no more than 5, given that data fluctuation is normally already compensated within 5 s. With $n$ to be 5 and $m$ to be 1, the average of the target point was mostly contributed from neighboring data points within about 11 seconds, according to a weight of 80%. This consequently led to a vertical resolution of about 22 m for the ascent and 11 m for the descent after smoothing. Similarly, the vertical resolution was about 50~60 m for the ascent and 25~30 m for the descent when $m$ was set to be 5. Different choices of $m$ gave a similar pattern of vertical profiles, but with some differences in details. In this study, $m$ was set to be 5 to achieve a better smoothing for further calculations, though this caused a loss of the vertical resolution more than twice as large as that when just smoothing once. A comparison was made between unsmoothed data, smoothed data using the FMS approach in this study and the ONA method in Hagler et al. (2011), as well as 20-m averaged data using those two algorithms. It was found that both algorithms could properly deal with data fluctuation caused by instrumental noises without introducing artificial features (e.g., Fig. 3a-3c). However, the FMS method seemed to be more capable of reliably removing outliers in some cases (e.g., Fig. 3d-3f). The comparison indicated that the FMS procedure could effectively reduce data fluctuation while still preserve reasonable variability of the profile."

Moreover, we have corrected a mistake in the description of the smoothing algorithm.

**P5, L8,** "…, where $i=1,2,…,N-1,N$ and $j=1,2,…,n-1,n$;"

**Section 2.2.2, page 5, line 8: "Details of the correction scheme developed for tackling with artifacts of AE-31 were described in Ran et al. (2016)". Note that Ran et al. (2016) is just a submitted paper. In the reference list the journal to which Ran et al. paper was submitted is missing. Please add it. Moreover, as the AE31 data could significantly change in function of the chosen correction function it is necessary to resume here at least the main points of the correction scheme adopted in Ran et al. as the paper is not yet available to the scientific community. With respect to this, depending on the chosen correction scheme (i.e. C factors for each wavelength of the AE31), the obtained angstrom exponent should change introducing an error on the retrieved $\sigma_{MAAP,880nm}$. A quantitative assessment of the variability of $\sigma_{MAAP,880nm}$ depending on the chosen correction scheme for the AE31 is called for. Moreover, I strongly recommend an analysis of the error propagation of $\sigma_{MAAP,880nm}$ on the obtained C for the AE51. As a matter of fact the C factor of 2.52 is reported here without any statistical treatment of its uncertainty. Finally, no reference was made to the C value of 2.05±0.03 for the AE51 reported in Ferrero et al. (2011a). It should very interesting to discuss the difference on the two C values in terms of the chemical composition of the aerosol in the NCP with respect to the Europe.**

We thank the referee for the helpful and instructive comments. We agree with the referee that more details should be given on the correction scheme adopted in Ran et al. (2016), which has just been published in Atmospheric Environment.

Briefly, the correction scheme used to deal with instrumental artifacts of AE-31 was a combination of the

modified Virkkula method (Virkkula et al., 2007) to treat the shadowing effect and the Schmid method (Schmid et al., 2006) to treat filter multiple scattering and aerosol scattering effects. The modified Virkkula method assumed a linear relationship between BC mass concentrations and time across the filter change, particularly, a quadratic relationship for special cases where ambient BC experienced a peak-shaped variation, instead of the assumption of constant BC mass concentrations in Virkkula et al. (2007). Following procedures in the Schmid method, the wavelength-dependent correction factor ($C_\lambda$) could be derived. As the referee stated, the choice of the correction scheme for AE-31 measurements might introduce uncertainties to absorption Angström exponent ($\alpha$), and thereby pass them to $\sigma_{MAAP,880nm}$ that were calculated from $\alpha$ and absorption coefficients measured at 637 nm ($\sigma_{MAAP,637nm}$) by MAAP. Using a constant $C$ factor for AE-31 as also often used in some studies (e.g., Weingartner et al., 2003; Sandradewi et al., 2008) instead of the wavelength-dependent $C_\lambda$ results in an underestimation of $\alpha$ over the 660-880 nm spectrum by about 19.5%. This consequently leads to an overestimation of $\sigma_{MAAP,880nm}$ and the $C$ factor for AE-51 by about 9.6% and 8.4%, respectively.

Another important thing to mention is that the actual wavelength of MAAP is 637 nm instead of the nominal wavelength of 670 nm (Müller et al., 2011), as pointed out by one of the referees for Ran et al. (2016). We have accordingly corrected all related results in the revised manuscript. Subsequently, attenuation coefficients $\sigma_{AE-51,880nm}$ (ATN<10) measured by AE-51 and calculated $\sigma_{MAAP,880nm}$ were employed to yield the $C$ factor using reduced major axis regression (Fig. R3, also as Fig. 2 in the revised manuscript). The $C$ factor was 2.98±0.05 with 95% confidence, quite different from a value of 2.05±0.03 in Ferrero et al. (2011a). Possible explanations on such a difference in the $C$ factor might be found in aerosol chemical compositions in the NCP region and the Po Valley basin. Besides, the $C$ factor in Ferrero et al. (2011a) was obtained from Mie calculations, and thus was subject to uncertainties resulting from assumptions such as BC size distributions, BC mixing state and particle morphology. Also the $C$ factor derived from methods in this study bears some uncertainties as mentioned above.

[Figure]

Figure R3. Reduced major axis regression of attenuation coefficients $\sigma_{AE-51,880nm}$ (ATN<10) measured by AE-51 and absorption coefficients $\sigma_{MAAP,880nm}$ calculated from concomitant MAAP and AE-31 measurements in the comparative test.

Accordingly, revisions have been made in the manuscript as:

**P6, L11,** "AE-31 suffered instrumental artifacts in the same way as AE-51. Details of the correction scheme developed for tackling with AE-31 artifacts were described in Ran et al. (2016). Briefly, the correction scheme combined the modified Virkkula method (Virkkula et al., 2007) to treat the shadowing effect and the Schmid method (Schmid et al., 2006) to treat filter multiple scattering and aerosol scattering effects. The modified Virkkula method assumed a linear relationship of BC mass concentrations and time across the filter change, particularly, a quadratic relationship for special cases where ambient BC experienced a peak-shaped variation, instead of constant BC mass concentrations as in Virkkula et al. (2007). The wavelength-dependent correction factor ($C_\lambda$) could be obtained following procedures in Schmid et al. (2006). The temporal resolution of AE-31 during the comparative test was 2 min."

**P6, L30,** "Three steps were taken to obtain the $C$ factor. Firstly, aerosol absorption Angström exponent ($\alpha$) over the spectrum span of 660 and 880 nm was derived from absorption coefficients $\sigma_{AE-31,660nm}$ and $\sigma_{AE-31,880nm}$, which were corrected from attenuation coefficients at 660 and 880 nm measured by AE-31. Hence, $\alpha$ was calculated from:

$$\alpha = \frac{\ln(\sigma_{AE-31,660nm}) - \ln(\sigma_{AE-31,880nm})}{\ln(880) - \ln(660)} \ . \tag{2}$$

Secondly, $\alpha$ for the spectrum of 660 and 880 nm was used to represent $\alpha$ over the span of 637 and 880 nm. Therefore, $\sigma_{MAAP,880nm}$ was quantified from measured $\sigma_{MAAP,637nm}$ following the spectral dependence of aerosol absorption coefficients in the form of $\lambda^{-\alpha}$:

$$\sigma_{MAAP,880nm} = \sigma_{MAAP,637nm} \times (\frac{880}{637})^{-\alpha} \ \ , \tag{3}$$

Finally, reduced major axis regression of attenuation coefficients $\sigma_{AE-51,880nm}$ (ATN<10) measured by AE-51 and absorption coefficients $\sigma_{MAAP,880nm}$ calculated from MAAP and AE-31 yielded the $C$ factor of 2.98±0.05 with 95% confidence (Fig. 2). It was noted that the $C$ factor for AE-51 was reported as 2.05±0.03 with 95% confidence in Ferrero et al. (2011a). Possible explanations on such a difference in the $C$ factor might be found in aerosol chemical compositions in the NCP region and the Po Valley basin. Besides, the $C$ factor in Ferrero et al. (2011a) was obtained from Mie calculations, and thus was subject to uncertainties resulting from assumptions such as BC size distributions, BC mixing state and particle morphology. In addition, the choice of the correction scheme for AE-31 measurements in this study might introduce uncertainties to $\alpha$ and thereby $\sigma_{MAAP,880nm}$. Using a constant $C$ factor for AE-31 as also often used in some studies (e.g., Weingartner et al., 2003; Sandradewi et al., 2008) instead of the wavelength-dependent $C_\lambda$ results in an underestimation of $\alpha$ over the 660-880 nm spectrum by about 19.5%. This consequently leads to an overestimation of $\sigma_{MAAP,880nm}$ and the $C$ factor for AE-51 by about 9.6% and 8.4%, respectively."

**Page 5, lines 12-13: "Measured $\sigma_{AE-51,880nm}$ (ATN<10) and calculated $\sigma_{MAAP,880nm}$ were linearly fitted with a correlation coefficient of 0.96 in a significant level (P<0.001), yielding a C value of 2.52": Please add the picture of this correlation.**

We have added a figure as Fig. 2 in the revised manuscript to show this correlation. The figure is given in the response to the last comment.

**Page 6, line 13, equation 6: "$H_m$ was calculated from a sigmoid function that could well characterize typical daytime profile of $m_{BC}$:". From this sentence it appears that $H_m$ was calculated using equation 6. However, equation 6 requires as input both the mixing layer and the entrainment layer. This point is not clearly defined and needs to be specified. I also suggest to add a graphical example of the mixing layer determination using the sigmoid function. Finally a question: as you have both the potential temperature and wind profiles at disposal, have you ever thought to analyse the mixing layer also using the Richardson**

**number approach?**

We thank the referee for pointing out the interpretation that might have caused confusion. We have added an illustration to show the fitting of BC profiles using the sigmoid function (Fig. R4, also as Fig. S4 in the supplement). We have clarified this point in the revised manuscript as:

**P8, L24,** "On the other hand, typical daytime profiles of $m_{BC}$ could be well characterized by the sigmoid function:

$$m_{BC} = C_{ms} - \frac{C_{ms} - C_{fs}}{e^{-(h - H_{m,BC,sigmoid})/H_e} + 1} \qquad (6)$$

where $C_{ms}$ and $C_{fs}$ are respectively characteristic $m_{BC}$ within the ML and in free troposphere (FT), $H_{m,BC,sigmoid}$ is the mixing height derived from BC vertical profiles using the sigmoid function, $H_e$ represents the thickness of the EL, $h$ is the height at which each 20-m averaged $m_{BC}$ is obtained. The parameters $C_{ms}$, $C_{fs}$, $H_{m,BC,sigmoid}$ and $H_e$ could be directly determined by fitting measured $m_{BC}$ at each height $h$ using Eq. (6) as shown by the example (Fig. S4). A comparison was made between $H_{m,BC,gradient}$ and $H_{m,BC,sigmoid}$ for typical daytime BC vertical profiles. Results from the two methods agreed quite well with each other, with a difference of less than 2 % (Fig. S5). In addition to reliably estimating the mixing height as the gradient method, the sigmoid function could also directly determine parameters including $C_{ms}$, $C_{fs}$, and $H_e$. Therefore, the sigmoid function was chosen to obtain all parameters for typical daytime BC profiles."

[Figure]

Figure R4. An example of fitting BC vertical profiles using the sigmoid function. Measurements were conducted on July 8, 2014 (10:41-11:27 LT).

Finally, we followed the referee's suggestion and employed the Richardson number approach to determine the mixing height (Vogelezang and Holtslag, 1996; Seibert et al., 2000). Equation (R1) was used to calculate the Richardson number $Ri_b(h)$ for each 5-m layer at the midpoint $h$, where $\theta_v(h)$ is the virtual potential temperature calculated from potential temperature and the mixing ratio of water vapor, $\theta_{v1}$ is the average virtual potential

temperature for the 5-10 m layer, $U(h)$ and $V(h)$ are wind components computed from wind speed and direction, $g$ is the gravity of earth. The mixing height was determined as the height where $Ri_b(h)$ exceeded the classic critical value of 0.25 (Seibert et al., 2000).

$$Ri_b(h) = \frac{gh}{\theta_{v1}} \frac{\theta_{v1}(h) - \theta_{v1}}{U(h)^2 + V(h)^2}$$  (R1)

Figure R5 shows a satisfactory agreement among mixing heights estimated from vertical profiles of $\theta$ ($H_{m,\theta}$) and $q$ ($H_{m,q}$) using the gradient method and from the Richardson number approach ($H_{m,RN}$). However, uncertainties in the determination of mixing heights using the Richardson number approach might arise from the accuracy of temperature and wind profiles, the choice of the equation and the critical value. Moreover, the height of the nocturnal boundary layer was poorly determined and corresponding results have been removed in Fig. R5. Therefore, a combination of the sigmoid approach and the gradient method was applied to estimate mixing heights for the entire dataset.

[Figure]

Figure R5. A comparison among mixing heights estimated from vertical profiles of $\theta$ ($H_{m,\theta}$) and $q$ ($H_{m,q}$) using the gradient method, and that from the Richardson number approach ($H_{m,RN}$).

**MINOR POINTS: Page 7, lines 3-4: "the normalized height ($H_{Nor}$), which was calculated from $h/H_m$-1". In Ferrero et al. (2014) this analysis is explained. Add this reference at the end of the sentence.**

We thank the referee for this helpful comment. We have revised the manuscript as:

**P9, L17,** "Statistically, vertical profiles of BC were categorized into two types, according to their shapes along the normalized height ($H_{Nor}$), which was calculated from $h/H_m$-1 (Ferrero et al., 2014)."

**Figure 2b: at $H_{Nor}$=0 BC data are characterized by free troposphere concentration levels. I was a bit surprised about it. I expected that around $H_{Nor}$=0 there was at least the end of the exponential decrease of concentration starting from ground values. Could you comment it?**

We thank the referee for pointing this out. As expected, the height around $H_{\text{Nor}}=0$ is indeed the end of the exponential decrease of $m_{\text{BC}}$ starting from the ground value for individual profile. Figure 2b (as Fig. 5b in the revised manuscript) displays each vertical profile of $m_{\text{BC}}$ in the evening (grey lines). Above the NBL ($H_{\text{Nor}}>0$), no apparent decrease in $m_{\text{BC}}$ was found for individual profile. However, the level of $m_{\text{BC}}$ above $H_{\text{Nor}}=0$ differed largely in different cases, representing clean or relatively polluted conditions in FT. Hence, the average profile presents an artificial feature of a decrease in $m_{\text{BC}}$ even above $H_{\text{Nor}}=0$. To clarify the confusing feature of the average profile, we have revised the manuscript as:

**P9, L27,** "For each BC profile (grey lines in Fig. 5b), $m_{\text{BC}}$ nearly exponentially declined with $H_{\text{Nor}}$, as a result of weakened turbulence and vertical dispersion."

**Page 3, lines 24-25 and equation 1: "to estimate aerosol absorption coefficients at the wavelength of 880 nm following"… Please note that $\sigma_{\text{AE-51,880nm}}$ is the attenuation coefficient and not the absorption coefficient as reported in many papers (i.e. starting from Weingartner et al. (2003)). Please correct the paper for this point.**

We thank the referee for this valuable comment. We have corrected the manuscript as:

**P4, L18,** "…simultaneously detected to obtain attenuation coefficients at the wavelength of 880 nm $(\sigma_{\text{AE-51,880nm}})$…"

[revised manuscript text omitted]

